# Performance Scaling via Optimal Transport: Enabling Data Selection from Partially Revealed Sources

**Feiyang Kang**[*]
Bradley Department of ECE
Virginia Tech
Blacksburg, VA 24061
`fyk@vt.edu`

**Hoang Anh Just**[*]
Bradley Department of ECE
Virginia Tech
Blacksburg, VA 24061
`just@vt.edu`

**Anit Kumar Sahu**
Alexa AI
Amazon
Seattle, WA 98121
`anit.sahu@gmail.com`

**Ruoxi Jia**
Bradley Department of ECE
Virginia Tech
Blacksburg, VA 24061
`ruoxijia@vt.edu`

## Abstract

Traditionally, data selection has been studied in settings where all samples from prospective sources are fully revealed to a machine learning developer. However, in practical data exchange scenarios, data providers often reveal only a limited subset of samples before an acquisition decision is made. Recently, there have been efforts to fit scaling functions that predict model performance at any *size and data source composition* using the limited available samples. However, these scaling functions are usually black-box, computationally expensive to fit, highly susceptible to overfitting, or/and difficult to optimize for data selection. This paper proposes a framework called `projektor`, which predicts model performance and supports data selection decisions based on partial samples of prospective data sources. Our approach distinguishes itself from existing work by introducing a novel *two-stage* performance inference process. In the first stage, we leverage the Optimal Transport distance to predict the model's performance for any data mixture ratio within the range of disclosed data sizes. In the second stage, we extrapolate the performance to larger undisclosed data sizes based on a novel parameter-free mapping technique inspired by neural scaling laws. We further derive an efficient gradient-based method to select data sources based on the projected model performance. Evaluation over a diverse range of applications (e.g., vision, text, fine-tuning, noisy data sources, etc.) demonstrates that `projektor` significantly improves existing performance scaling approaches in terms of both the accuracy of performance inference and the computation costs associated with constructing the performance predictor. Also, `projektor` outperforms by a wide margin in data selection effectiveness compared to a range of other off-the-shelf solutions. We provide `projektor` as an open-source toolkit.

## 1   Introduction

The choice of training data is one of the most crucial components when it comes to extracting the best performance out of a model. Since data is typically acquired from various sources, such as different organizations or vendors, machine learning practitioners often encounter a central question: *how to select and combine samples from these data sources?*

---

[*]Equal contribution. This work also appears at *Data-centric Machine Learning Research (DMLR)* Workshop@ICML 2023. Code Repository: https://github.com/ruoxi-jia-group/projektor.

37th Conference on Neural Information Processing Systems (NeurIPS 2023).

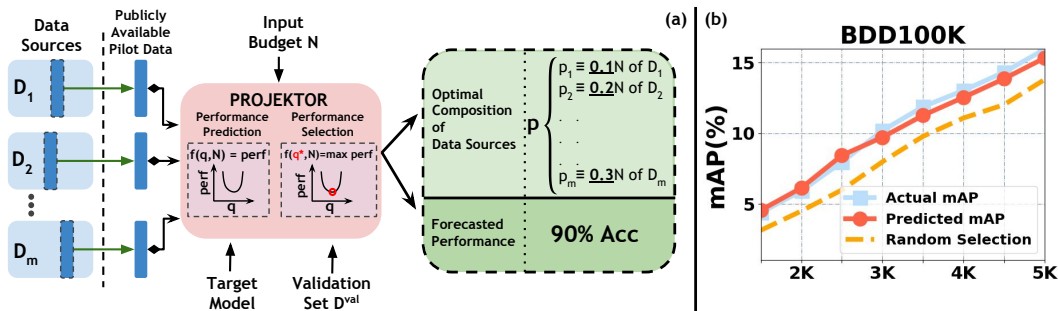

Figure 1: (a) Overview of `projektor`, which take as inputs the public pilot data from each source, a selection budget, a target model, a validation set representing the test distribution, and return the optimal combination of data sources as well as the prediction of the resulting model performance. (b) Optimal data source composition and performance projection in a practical autonomous driving data acquisition scenario [15]. `projektor` achieves accurate model performance projection from 1K pilot samples and effective selection. Please see Evaluation Metrics (Section 5) for details on mAP.

Although data selection has been extensively studied in the literature related to active learning [1], coreset selection [2], and data valuation [3–8], most techniques are designed for a *fully-observable* setting where all data sources are fully revealed to the model developer. The core ideas behind these techniques are to compare the relative importance of different data points or enumerate possible combinations of data points, all of which require complete knowledge of the entire collection of data points. While these methods have shown promising results, their practical applications in real-world scenarios are limited due to a significant gap: the acquisition decision-making processes require knowledge of the entire data sets, while data owners may only reveal limited samples before an acquisition decision is made (e.g., [9, 10] provide the examples in real-world data markets).

To bridge the gap, this paper explores strategic data selection in *partially observable settings*, where only limited samples of data sources (referred to as pilot datasets) are accessible. The goal is to determine an optimal allocation of the selection budget to each source, only based on the pilot datasets, such that the model trained on the mixture of collected data achieves the best performance at some given objectives.

**Technical challenges.** In the fully-observable setting, the evaluation and eventually ranking the candidate data selection decisions, including the number of samples to be selected and the ratio of samples from each source (*"mixing ratios"*), can be determined directly on the available datasets [11, 12]. However, the partially observable setting presents considerable challenges for evaluating a selection decision as one can no longer directly evaluate model performance on the entire data. With limited samples from each data source, the best possible evaluation is the resulting model performance for any combination of the pilot datasets. Then, to make an informed selection decision, it is necessary to understand the model's performance when trained on potentially larger datasets (target scales) at various mixing ratios. In other words, there is a need for prediction and projection of model performance onto larger data scales at different mixing ratios.

A recent study [13] proposes a performance scaling law that takes into account the data size and mixing ratio to predict model performance. Providing a preliminary exploration of this problem, though, this approach faces two major limitations: (1) The numerical instability of its high-order form for scaling functions causes significant difficulty in fitting its parameters, rendering the fitted function susceptible to overfitting and often fails to extrapolate model performance on unseen data mixtures. (2) It hypothesizes on the *separability* of model performance scaling with data composition and with data size, which is generally untrue as evidenced by latest research [14] and leads to unsatisfactory performance prediction results in empirical observations. Besides, this method requires parameters that grow quadratically with the number of data sources, demanding many (mixing ratio, resulting performance) pairs to fit the function, resulting in substantial computational overhead. Thus, there remains a considerable lack of effective and practical approaches to this problem.

**Contributions.** The paper investigates two fundamental building blocks for strategic data selection in the partially observable setting:

(Q1) *How to provide an accurate projection of model performance trained on any combination of data sources based on limited samples?* and (Q2) *How to determine the optimal selection strategy?*

Towards that end, the paper makes the following contributions.

• *Parameter-Efficient Performance Prediction based on Optimal Transport (Addressing Q1).* In contrast to existing model performance scaling methods that feature a *one-shot* fitting of a non-informative parametric model ("*surrogate*"), our approach is a novel *two-stage* performance inference process, where the first stage addresses the dependence of the model performance on the mixing ratio by fitting a parameter-efficient model between model performance and Optimal Transport [16] distance between the mixtures of training data and the validation data (Section 4.2). Then, for stage two, we propose a parameter-free mapping that directly projects model performance onto larger data scales, achieving remarkable accuracy as it fully preserves this the dependency of model performance scaling with data sizes and data distributions (Section 4.3).

• *Determining optimal data selection strategies (Addressing Q2).* We consider the typical data selection goal: maximizing the resulting model performance with fixed data acquisition budgets (data quantity). With model performance predicted by the proposed tools, these problems translate into convex losses that are optimized effectively via gradient-based methods (Section 4.4). We also provide in Appendix.B how it similarly applies to alternative objectives such as minimizing data acquisition costs for the resulting model performance to reach a given level.

• *Experiments.* We experiment on a variety of applications (*vision*, *natural language processing (NLP)* etc., with simple to complex models) with a rich diversity of tasks and real-world scenarios (e.g., autonomous driving, fine-tuning, noisy data sources, unlabeled data). The proposed approach is highly effective in performance prediction, demonstrating superior prediction precision to many baselines while being much more efficient to be constructed (Section 5). We test the performance of data selection by optimizing the performance predictor and show that it improves over existing methods by $3\%$ on ImageNet-100.

## 2   Related Work

The recent line of research on **Data valuation** aims to assess the relative importance of each data source ("*value'*") to machine learning applications [3–8]. While originally designed for data pricing, these values are frequently used to inform data selection [4, 17]: in more detail, one can rank the data sources based on their values and select the data points with the highest values. While value-based selection shows some promising results, data values are not directly related to model performance and hence cannot inform the prediction of model performance resulting from the selected data. Besides, values for different data sources typically cannot be combined to measure the value of their compositions [18, 8]. Notably, distributional distances including Optimal Transport have seen a major presence in data valuation as an implicit proxy for model performance [8, 19], but no connection has been made to directly relate data distance to model performance. Our work bridges this gap and directly addresses this long-standing problem. On another line, **Coreset selection** attempts to find a representative subset of data points to speed up learning [2]. Coreset selection methods have been studied for different learning algorithms [20, 21]. For example, a popular coreset selection approach for neural networks is to cast data selection as a bilevel optimization problem that selects the optimal subset to maximize the performance evaluated on a validation dataset [22]. However, coreset selection techniques rely on access to all the data samples to be chosen from, which limits their use in the partially observable setting. Besides, **Predicting** the resulting **model performance** associated with a dataset without performing actual training on it has attracted a lot of attention in different use cases, such as interpreting the model learning process [23, 24]–which leverages surrogate functions to model the black-box relationships between model performance and training data, or predicting performance under the distributional shift [25]. Our work resembles the idea of predicting model performance from data but differs in the technique of leveraging the data distance in the performance predictor. **Scaling laws**, predicting how the model performance changes with the scale of training data, model parameters, and computation budget [26], have seen increasingly successful in a variety of tasks pertaining to vision and text processing [27]. The performance of machine-learning models generally adheres to a power law relationship with the scale of these variables, which allows for predicting the model performance on larger scales with high precision [26] and provides a viable approach to predicting the potential usefulness of target data from only a small proportion of the set. [14] shows that data from different distributions generally scale at different rates. Our work provides a novel approach that materializes this dependency of scaling relationships with data distributions and achieves remarkable empirical results.

# 3 Problem Formulation

**Data provider.** Suppose that there are $m$ prospective data providers. Datasets (*data sources*) held by these providers are denoted by $D_1^{\text{all}}, \ldots, D_m^{\text{all}}$, respectively. We focus on the case that only partial data (*samples*) from these sources are made available to the public, replicating practical data exchange scenarios [9]. We refer to the public subset of each data source as *a pilot dataset* and denote it by $D_i^{\text{pi}}$, where $D_i^{\text{pi}} \subseteq D_i^{\text{all}}$ and $|D_i^{\text{pi}}| = n_i \ll \bar{N}_i = |D_i^{\text{all}}|$ for all $i$. Each provider $i$, upon accepting the *purchasing order* for acquiring $n_i$ samples ($n_i \leq \bar{N}_i$), will randomly sample a subset $S_i$ of size $n_i$ from $D_i^{\text{all}}$ and return the subset to the requester.[2]

**Data collector (or requester, machine learning practitioner).** Now, consider a data collector who would like to acquire samples from the providers to train a model. Notably, *the collector's acquisition decisions must be made based only on the pilot datasets*. We assume the collector has a validation set $D^{\text{val}}$, representing the desired target data distribution. For ease of exposition, we assume the collector has a target learning algorithm $\mathcal{A}$ [3] that is going to be applied to the collected data as well as a target performance metric $\mathcal{L}$ which takes the input of a trained model and a validation set and returns a performance score. The model performance resulting from training on any dataset $S$ can be thus expressed as $\mathcal{L}(\mathcal{A}(S), D^{\text{val}})$.

Given a selection budget of $N$ samples, a mixing ratio of data sources $\mathbf{p} = \{p_1, \ldots, p_m\}$ such that $\forall i, 0 \leq p_i \leq 1$ and $\sum_{i=1}^{m} p_i = 1$, and $m$ datasets $D_1, \ldots, D_m$ to be mixed, we denote the selected dataset by $\mathcal{D}(N, \mathbf{p}) = S_1 \cup \cdots \cup S_m$, where each $S_i$ is a random subset of $D_i^{\text{all}}$ and $|S_i| = p_i N$. Using these notations, we now describe the typical acquisition goals that can be accommodated by our approach:

- (Primary) *Fixed-budget selection for maximal performance:* The collector seeks to maximize the resulting model performance by strategically choosing the mixing ratio $\mathbf{p}$ of $m$ data sources at a *pre-specified* selection budget $N_s \leq \sum_{i=1}^{m} \bar{N}_i$. The objective can be formalized as $\max_{\mathbf{p}} \mathcal{L}(\mathcal{A}(\mathcal{D}(N_s, \mathbf{p})), D^{\text{val}})$.

- (Alternative) *Flexible-budget selection for reaching performance threshold with minimal costs:* The collector seeks to attain a target model performance $u^{\text{tar}}$ by choosing *both* the mixing ratio $\mathbf{p}$ as well as the selection budget $N$. More formally, the objective can be expressed as $\min_{N, \mathbf{p}} N$ s.t. $\max_{\mathbf{p}} \mathcal{L}(\mathcal{A}(\mathcal{D}(N, \mathbf{p})), D^{\text{val}}) \geq u^{\text{tar}}$.

The alternative objective can be treated as a direct extension of the primary, where one solves the *"performance maximization"* problem for different data quantities $N$ and performs a *line search* for minimal data quantity $N$ that meets the performance requirement. We defer to Appendix.B for its complete solution procedure due to the similarity.

**Design challenge and key idea.** The primary challenge is that the collector cannot access $D_1^{\text{all}}, \ldots, D_m^{\text{all}}$ for decision making and hence cannot directly evaluate the two optimization objectives for every $N$. Yet, as the pilot datasets are public, the collector can evaluate and observe the model performance associated with various mixtures of the pilot datasets for $N \in \{N : Np_i \leq |D_i^{\text{pi}}|, i = 1, \ldots, m\}$ project the evaluations onto larger data scales. Our high-level idea to tackle the challenge is to first predict the model performance associated with any mixture of prospective unrevealed data sources based on observations on pilot datasets and project the predictions onto different data scales using scaling laws, then determine the data selection strategy by optimizing the predicted performance at the target scales.

# 4 Methodology of `projektor` : prediction, projection, and selection

## 4.1 Preliminaries on Optimal Transport

*Optimal Transport (OT)* is a metric for measuring the discrepancy between probability distributions [16]. Compared to other measures such as the Kullback-Leibler Divergence [29] or Maxi-

---

[2]This paper will assume each provider *honestly* provides requested samples and leave an in-depth study of potential security risks, such as malicious data manipulation [28] to future work.

[3]The proposed data selection approach can support multiple learning algorithms by simply applying it to different choices of algorithms/metrics and picking the best one.

mum Mean Discrepancies [30], OT enjoys advantageous analytical properties (is a valid metric; compatible with sparse-support distributions; stable with respect to deformations of the distributions' supports [31, 32]). Given probability measures $\mu_t, \mu_v$ over the space $\mathcal{Z}$, the OT distance is defined as [33] $\text{OT}(\mu_t, \mu_v) := \min_{\pi \in \Pi(\mu_t, \mu_v)} \int_{\mathcal{Z}^2} \mathcal{C}(z, z') d\pi(z, z')$, where $\Pi(\mu_t, \mu_v) := \left\{ \pi \in \mathcal{P}(\mathcal{Z} \times \mathcal{Z}) \mid \int_{\mathcal{Z}} \pi(z, z') dz = \mu_t, \ \int_{\mathcal{Z}} \pi(z, z') dz' = \mu_v \right\}$ denotes a collection of couplings between two distributions $\mu_t$ and $\mu_v$, $\mathcal{C} : \mathcal{Z} \times \mathcal{Z} \to \mathbb{R}^+$ is a symmetric positive-definite cost function (with $\mathcal{C}(z, z) = 0$), respectively. A popular choice of $\mathcal{C}$ is given in [34] by considering $z$ as the feature-label pair $(x, y)$. The computation of the OT distance usually relies on the Sinkhorn algorithm [35], which attains almost linear time complexity and memory overhead with the state-of-the-art implementations and applies to large scales with parallel computing [31, 32]. Given $D_t = \{(x_i, y_i)\}_{i=1}^N$ of size $N$, and $D_v = \{(x_i', y_i')\}_{i=1}^T$ of size $T$, one can construct discrete probability measures $\mu_t(x, y) := \frac{1}{N} \sum_{i=1}^N \delta_{(x_i, y_i)}$ and $\mu_v(x, y) := \frac{1}{T} \sum_{i=1}^T \delta_{(x_i', y_i')}$, where $\delta$ is the Dirac delta function. With slight abuse of notation, we use $\text{OT}(D_t, D_v)$ to denote the OT distance between their corresponding discrete measures $\text{OT}(\mu_t(x, y), \mu_v(x, y))$.

Extensive theoretical studies show that the OT distance between two distributions provides an upper bound on the difference of a model's performance when evaluated on the two distributions [36, 37, 8]. Largely built upon Kantorovich-Rubinstein Duality [38], existing theoretical results require assumptions on the Lipschitz constant of the model with respect to the input space. However, the constant is rarely known in practice, nor can it be bounded tightly for complex models such as deep neural networks. As a result, despite its widespread popularity as a performance proxy [8, 36, 39], one cannot directly apply the existing theoretical results to directly estimate the model performance based on the OT distance, posing an important gap.

### 4.2 Aligning data distance with performance predictions

Inspired by the theoretical results that the upper bound on the difference between training loss and validation loss can be tightly bounded by an affine transformation of the OT distance [33, 8], our first proposed approach is to directly estimate this transformation by empirically fitting data distances to model performance and then the estimated transformation can be used for predicting the model performance for different data mixtures. Formally, we consider the following performance estimator:

$$\hat{\mathcal{L}}\left(\mathcal{A}(\mathcal{D}(N, \mathbf{p})), D^{\text{val}}\right) = a_1 \cdot \text{OT}\left(\mathcal{D}(N, \mathbf{p}), D^{\text{val}}\right) + a_0, \tag{1}$$

where scaling parameter $a_1$ and centering parameter $a_0$ define the affine transformation. These two parameters can be estimated through least-square fitting. In particular, consider collecting the "training data" by forming the set of tuples $\{(N_j, \mathbf{p}_j, \mathcal{L}(\mathcal{A}(\mathcal{D}(N_j, \mathbf{p}_j)), D^{\text{val}}))\}_{j=1}^\ell$, where $N_j$ is randomly sampled from $\{1, \ldots, \sum_{i=1}^m |D_i^{\text{pi}}|\}$ and $\mathbf{p}_j$ is sampled from a probability simplex. Then, these parameters can be estimated as

$$(\hat{a_1}, \hat{a_0}) = \arg\min_{a_1, a_0} \sum_{j=1}^\ell \left(\hat{\mathcal{L}}\left(\mathcal{A}(\mathcal{D}(N_j, \mathbf{p}_j)), D^{\text{val}}\right) - \mathcal{L}\left(\mathcal{A}(\mathcal{D}(N_j, \mathbf{p}_j)), D^{\text{val}}\right)\right)^2.$$

We refer to this method as *center-scaling* (`projektor/CS`). $a_1$ can be considered an *empirical estimate of the Lipschitz constant*, and this treatment has been informally adopted in various works under different names [40–42]. `projektor/CS` has only two parameters that need to be estimated, which brings an important benefit of efficiency: we only need to have a few training iterations to get the "training data" for the above least-square fitting.

While proposed `projektor/CS` is sufficient to provide reliable performance predictions in most circumstances, we found it possible to further improve the prediction accuracy by making the scaling parameter and centering parameter a function of the mixing ratio. The intuition is that for samples from different data sources (i.e., data lying in different manifolds of the input space), the Lipschitz constant of the model along the combined manifold may vary with the mixing ratio. Hence, we supplement `projektor/CS` with simple nonlinear terms to characterize the dependence on each data source, leading to the *pseudo-quadratic* (`projektor/PQ`) method, which is given as

$$\hat{\mathcal{L}}(\mathcal{A}(\mathcal{D}(N, \mathbf{p})), D^{\text{val}}) = \sum_{i=1}^m (b_2^i \cdot p_i^2 + b_1^i \cdot p_i + b_0) \cdot \text{OT}(\mathcal{D}(N, \mathbf{p}), D^{\text{val}}) + \sum_{i=1}^m (c_2^i \cdot p_i^2 + c_1^i \cdot p_i + c_0), \tag{2}$$

where $(\mathbf{b_2}, \mathbf{b_1}, \mathbf{b_0}, \mathbf{c_2}, \mathbf{c_1}, \mathbf{c_0})$ are pseudo-quadratic parameters where the fitting process is similar to Eq. (1). `projektor/PQ` has $\mathcal{O}(m)$ parameters ($m$ is the number of data sources). So its fitting

process will require more re-training than `projektor/CS`. However, as we will show in Section 5, it still significantly improves the efficiency over the existing baselines [13]. This pseudo-quadratic form is chosen for it contains the simplest nonlinear terms and we want to preserve the convexity for numerical stability. We trimmed off the cross terms in the quadratic function as they often do not contribute much, resulting in the number of parameters growing linearly rather than quadratically with the number of data sources as in [13], greatly easing the computation burden in parameter fitting.

### 4.3 Parameter-free performance projection onto larger data scales

Once the parameters are learned, the performance predictors (1) and (2) can be used to predict the validation performance associated with a training set by calculating the OT distance between the training and the validation set and plugging it as input to the predictors. Then, we need to project these predictions onto the target data scales. Neural scaling laws showcase the predictability of empirical performance with respect to the size of the training dataset, where it typically follows a log-linear scaling relationship as

$$\mathbb{E}_V[\mathcal{L}(\mathcal{A}(\mathcal{D}(N, \mathbf{p})); D^{\text{val}})] \approx -\alpha \log(N) + C$$

where $\alpha$ and $C$ are some constants [26]. Recent work [14] shows that when data from different sources differ in "quality" (e.g., noise level), which is the most likely scenario in practice, the scaling parameters are often vastly different, rendering parameters $\alpha$ and $C$ *functions* of data composition $\mathbf{p}$ and model performance for different data mixtures $\mathbf{p}$ scales with different rates. [13] assumes the same constant $\alpha$ for all data mixtures, leading to unsatisfactory scaling results. The difficulty underlying this dependence is that these scaling parameters differ for every data mixture and there is no closed-form expression available for this functional relationship. With the performance prediction tools proposed above, it is possible to directly predict model performance of any data mixture at the scale the tool has been fitted. That is, for any given data scale $N_0$, as long as one completes the one-off fitting process of the performance predictor, model performance for any data composition at data size $N_0$ can be inferred directly using Eq. (1) or Eq. (2). Thus, by performing the fitting process at different small scales for once, for any desired data mixture, we can directly fit the neural scaling laws for this particular distribution and project it onto larger data scales, without needing to train any additional model or make any further approximations. We formalize it into the following theorem.

**Theorem 1** (Data Composition Dependent Performance Projection)**.** *Consider log-linear performance scaling relationship depending on both data size $N$ and data composition $\mathbf{p}$ given as $\mathbb{E}_V[\mathcal{L}(\mathcal{A}(\mathcal{D}(N, \mathbf{p})); D^{val})] = -\alpha(\mathbf{p})\log(N) + C(\mathbf{p})$. Assume one has completed the fitting of the performance predictor on two different scales $N_0 < N_1$, which gives $\hat{\mathcal{L}}(\mathcal{A}(\mathcal{D}(N_0, \mathbf{p})); D^{val})$ and $\hat{\mathcal{L}}(\mathcal{A}(\mathcal{D}(N_1, \mathbf{p})); D^{val})$ for all data mixtures $\mathbf{p}$. Then, the model performance $\hat{\mathcal{L}}(\mathcal{A}(\mathcal{D}(N, \mathbf{p})); D^{val})$ for any data mixture $\mathbf{p}$ at any data scale $N$ can be predicted as*

$$\hat{\mathcal{L}}(\mathcal{A}(\mathcal{D}(N, \mathbf{p})); D^{val}) = \left(\log \frac{N_1}{N_0}\right)^{-1} \left[\log \frac{N}{N_0} \hat{\mathcal{L}}(\mathcal{A}(\mathcal{D}(N_1, \mathbf{p})); D^{val}) - \log \frac{N}{N_1} \hat{\mathcal{L}}(\mathcal{A}(\mathcal{D}(N_0, \mathbf{p}); D^{val}))\right]$$

(3)

*without requiring fitting any additional parameters.* The proof and derivations are given in Appendix.B. We refer to this method as *parameter-free projection* for model performance. As this procedure does not rely on any additional assumption or parameterized surrogate, it requires minimal computational overhead while achieving considerably higher prediction accuracy compared to existing approaches such as [13]. Not exclusive to the performance predictor proposed in this work, this method can be plugged into other predictors seamlessly and provides benefits at large, marking a novel contribution to performance projection in data acquisition problems.

### 4.4 Performance-guided data selection

The intention of creating the proposed tools is not limited to providing predictions for model performance, rather, we expect the predictions to support determining the optimal data acquisition strategy. We show that these problems are convex and differentiable (Appendix.B) and thus can be solved effectively via gradient-based methods. Specifically, for our primary objective *fixed-budget selection for maximal performance*, with the proposed performance predictors with projection, we solve for $\mathbf{p}^* = \arg\max_{\mathbf{p}} \hat{\mathcal{L}}(\mathcal{A}(\mathcal{D}(N_s, \mathbf{p})), D^{\text{val}})$. We solve it iteratively with the following procedure. First, we initialize the algorithm with $\mathbf{p} = \mathbf{p^0}$ where $\mathbf{p^0}$ can be chosen arbitrarily provided that

$\sum_i p_i = 1$. Then, at each step, we perform the gradient update as

$$\mathbf{p^{t+1}} \leftarrow \mathbf{p^t} + d^t \cdot \left.\frac{\partial \hat{\mathcal{L}}(\mathcal{A}(\mathcal{D}(N_s, \mathbf{p})), D^{\text{val}})}{\partial \mathbf{p}}\right|_{\mathbf{p}=\mathbf{p}^t}$$

where $d^t$ is the step size at iteration $t$ and we obtain $\mathbf{p}^* = \mathbf{p^T}$ at convergence as the desired solution. Optimal Transport naturally provides its gradient w.r.t. the probability mass of data points in its dual solutions [8], which directly gives the gradients w.r.t. data mixtures $\mathbf{p}$. This easy availability of gradients renders the optimization highly efficient in computation, resulting in remarkably fast solutions. We use the *calibrated gradient* of OT from [8] which ensures the updated mixture $\mathbf{p}$ remains within the simplex $\sum_i p_i = 1$ at each step. We provide technical details of the gradient computation in Appendix.B. The alternative objective can be treated as a direct extension of the primary and we defer to Appendix.B for its solution procedure. The pseudo-code for `projektor` is provided in Appendix A.

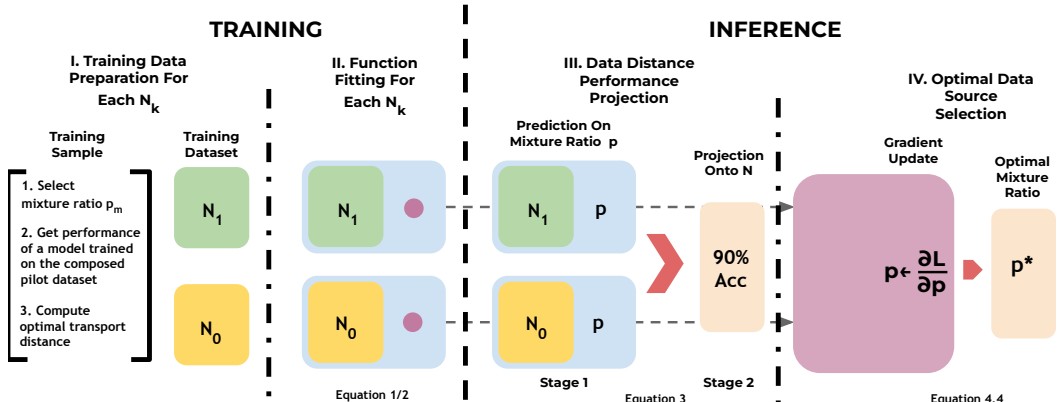

Figure 2: Workflow chart for the *two-stage* performance inference process of `projektor`.

## 5 Evaluation

In this section, we cover two main applications for `projektor`. **1)** *Performance projection*, where for any mixing ratio of data sources and any data scale, we want to predict the performance of the model trained on a given composed dataset. We also demonstrate efficiency and efficacy of our method for different scenarios of data sources, such as mislabeled or unlabeled data. **2)** *Optimal data source acquisition strategy*, where for a given data budget, we find a mixing ratio of data sources that can maximize the performance of a model. We present a solution to select optimal data source composition for two learning paradigms scenarios: training from scratch and model fine-tuning.

We compare with six existing baseline methods, where the first four: (1) Linear [13], which assumes a linear relationship with the data compositions; (2) Pseudo-Quadratic assumes a simple non-linear relationship; (3) Quadratic [13] assumes a fully quadratic relationship; (4) Rational [13] models a relationship through the sum of a set of rational functions. For $m$ data sources, each function has $m$ parameters, and there are m such functions, totaling $m^2$ parameters; (5) LOO [5] measures the importance of a data source by computing the performance difference after removing that source; (6) Shapley [18] is a game-theoretic method which computes the average marginal contribution of a data source to different subsets of other sources. Baselines (5) and (6) are suitable for informing the selection of data sources but are unable to predict model performance, so we only include them in data source selection experiments. Details on implemented baselines are described in Appendix D and further explained in [13]. For all experiments, we set up the problem with three data sources, where each source consists of different classes, and we refer the reader for additional information on the experimental setup, algorithm, datasets, models, implementations, code repository, and ablation studies on the number of data sources to Appendix D. We also showcase the runtime vs performance prediction trade-off and comparison with baseline methods.

**Evaluation Metrics.** We use mean absolute error (**MAE**) to measure the performance of our method by calculating the absolute difference between the predicted accuracy and the actual accuracy. For

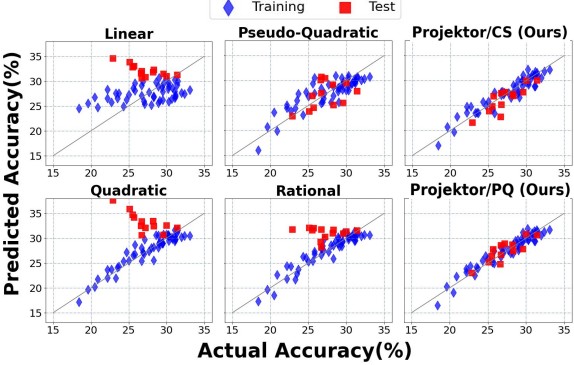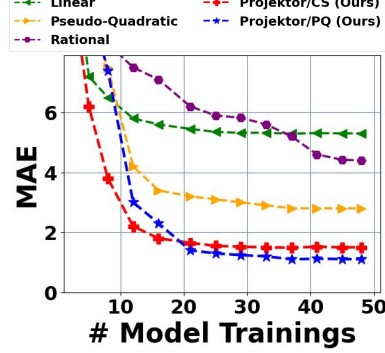

Figure 3: Predicted model performance vs. actual model performance for extrapolation on CIFAR-10 with total of 1000 data points for 3 data sources. Comparison between `projektor` and baselines.

Figure 4: MAE vs. number of model trainings used for fitting the method. Efficiency comparison between `projektor` and baselines.

the object detection task, we adopt a commonly used metric, mean average precision (**mAP**), which measures the average precision of a model across multiple classes or categories, providing a single value that represents the overall accuracy. The average precision represents the area under the precision-recall curve for a single class.

**Hyperparameters.** For practical reasons, we set the data scale $N_1$ in Eq. 5 to be the size of the smallest pilot dataset, i.e. $N_1 = \min_i |D_i^{\mathrm{pi}}|$. Upon selecting $N_1$, we empirically choose $N_0$ to be $\frac{2}{3} N_1$. For further investigations on selecting $N_0$, we provide sensitivity analysis on $N_0$ in Appendix D.

## 5.1 Performance Prediction

| Method | MNIST | CIFAR-10 | ImageNet100 | IMDB |
|---|---|---|---|---|
| Projektor/PQ (Ours) | $\underline{1.63} \cdot 7.27 (\triangle\ 3.01)$ | $\underline{0.84} \cdot \mathbf{1.26}(\triangle\ \mathbf{0.00})$ | $\underline{0.53} \cdot \mathbf{0.38}(\triangle\ \mathbf{0.00})$ | $\underline{0.25} \cdot \mathbf{1.16}(\triangle\ \mathbf{0.00})$ |
| Projektor/CS (Ours) | $3.30 \cdot \mathbf{4.26}(\triangle\ \mathbf{0.00})$ | $0.97 \cdot 1.95(\triangle\ 0.69)$ | $0.54 \cdot 1.03(\triangle\ 0.65)$ | $0.90 \cdot 1.87(\triangle\ 0.71)$ |
| Linear | $5.27 \cdot 28.03(\triangle\ 23.77)$ | $4.44 \cdot 6.04(\triangle\ 4.78)$ | $1.47 \cdot 4.59(\triangle\ 4.21)$ | $1.88 \cdot 7.48(\triangle\ 6.32)$ [d] |
| Pseudo-Quadratic | $\underline{1.63} \cdot 7.27(\triangle\ 3.01)$ | $1.82 \cdot 2.90(\triangle\ 1.64)$ | $0.77 \cdot 2.72(\triangle\ 2.34)$ | $0.43 \cdot 2.74(\triangle\ 1.58)$ |
| Quadratic | $\underline{1.63} \cdot 7.27(\triangle\ 3.01)$ | $0.88 \cdot 5.22(\triangle\ 3.96)$ | $0.50 \cdot 2.34(\triangle\ 1.96)$ | $0.43 \cdot 2.74(\triangle\ 1.58)$ |
| Rational | $5.28 \cdot 27.29(\triangle\ 23.03)$ | $\underline{0.83} \cdot 2.95(\triangle\ 1.69)$ | $\underline{0.44} \cdot 2.27(\triangle\ 1.89)$ | $1.91 \cdot 7.11(\triangle\ 5.95)$ |

Table 1: Training and test MAE values for extrapolation of data source compositions for each method. For each cell the left/right column value denotes the MAE value for predicting the training/test data, respectively. Underlined green/**bold red** denotes lowest training/test MAE value, respectively.

### A. Predicting Performance for Unseen Data Mixtures p

In this experiment, we fit the parameters on limited compositions and extrapolate the prediction to unseen compositions. Specifically, we choose one data source and limit its maximum composition to $< 55\%$ of contribution in the training set of the performance predictor, then we predict accuracy on the compositions consisting of $\geq 55\%$ of contribution. As we observe in Figure 3, Linear and Pseudo-Quadratic methods cannot fit well the training data, which indicates that these methods do not have a strong representation power. While Quadratic and Rational baselines can fit the training data, they suffer from overfitting and do not generalize to unseen compositions. On the other hand, as seen in Table 1, our method `projektor/PQ` achieves the best training and extrapolation performance. `projektor/CS` achieves second best extrapolation performance. Furthermore, we analyze the efficiency of our method compared to other baselines. As observed in Figure 4, `projektor` not only achieves the lowest MAE score but also converges with around 15 training data for `projektor/CS` and 25 for `projektor/PQ`, which demonstrates low computational requirement of our method. With shown strong predictive power of `projektor`, we now proceed to practical applications in performance projections onto larger data scales.

### B. Performance Projection to Larger Data Scales

**Mislabeled Data Sources**. In this experiment, we project performance onto larger data scales and also assume a more practical setting where data sources might not be of high quality and contain

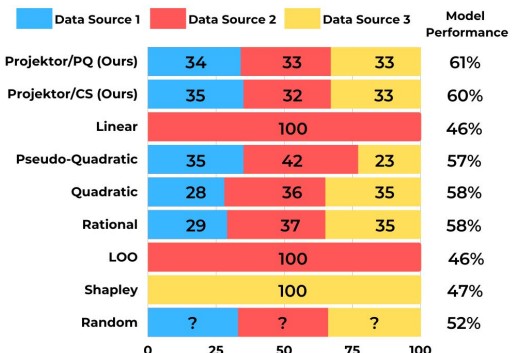
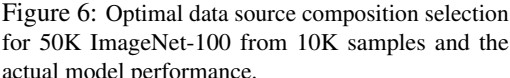

Figure 6: Optimal data source composition selection for 50K ImageNet-100 from 10K samples and the actual model performance.

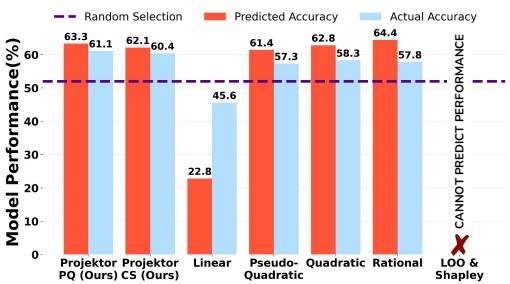

Figure 7: Performance projection of selected *optimal* mixture ratios in (Fig. 6) onto 50K ImageNet-100 from 10K samples. Comparison with the actual model performance.

noisy labels [43]. It is then critical to factor such irregularity for the performance prediction into our method. Given three mislabeled data sources formed by sampling CIFAR-10, each of which releases a pilot dataset of size 1K, we project performance for various mixing ratios onto larger data sizes, i.e. 2K, 5K, 7K, and 10K. We then measure the MAE value across all data scales. We observe in Fig. 5 that `projektor` achieves the best projection performance compared to all baseline methods. `projektor/PQ` achieves the lowest MAE score below $2\%$ and `projektor/CS` is slightly above $2\%$. The improved performance of our method can be attributed to the incorporation of actual data distance computation. This inclusion allows for a more accurate representation of mislabeling information in performance projection, unlike baseline methods that neglect this crucial information.

The promising results demonstrate the potential of our method to project performance of any composition to any data scale, which is important in the case of mislabeled data sources in the partially-revealed setting.

**Unlabeled Data Sources.** As mentioned earlier, data sources often contain noisy labels, and the process of labeling data can be costly. On the other hand, it is not uncommon to encounter unlabeled data sources. Therefore, we would like to extend our method to accommodate the setting of data sources without labeled data, and we aim to project performance of unlabeled data compositions from pilot data source mixtures of 1K samples from CIFAR-10. The three data sources contain unlabeled data from different classes of CIFAR-10. In this case, we compute the optimal transport distance on the feature space only, and we assume access to the labels of the pilot datasets,

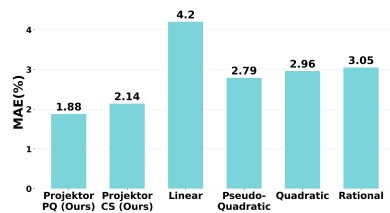

Figure 5: Performance projections from 1K CIFAR-10 samples across various mixing ratios and larger data scales: 2K, 5K, 7K, 10K. Comparison between `projektor` and baselines.

which enables us to train a model and obtain performance values. Consequently, we project performance across various mixing ratios onto larger data sizes (2K, 5K, 7K, and 10K). To visualize the performance of our method, we plot the difference between the projected performance and the actual performance. The closer the value approaches zero, the more optimal the projection becomes. Surprisingly, as we illustrate in Fig. 8, `projektor` can consistently maintain good projection performance for larger data sizes. Even at $10K$, the largest error is below $10\%$. These outcomes show that `projektor` can also be extended to unlabeled data sources, which demonstrates the flexibility and practicality of our method.

## 5.2 Optimal Data Source Selection

Training a model requires a tremendous amount of resources to tune hyperparameters to achieve the highest performance. We demonstrate that by choosing data strategically, we can also improve model performance. We consider a setting, where we are facing the problem of choosing only $50K$ to train ResNet-50 on ImageNet100 and would like to maximize the model's performance.

However, we are provided with only a pilot dataset of size $10K$ from each data source. As we observe in Figure 6, our optimized mixing ratio based on (4.4) achieves the highest model performance compared to all baselines. Further, using the functions from the previous step, we project the performance of our selected mixing ratio, and we observe in Figure 7 that `projektor` not only most closely predicts the accuracy to the actual accuracy but also attains the highest actual accuracy out of all methods. The improved selection of mixture ratios in our method can be attributed to our proposed selection approach (4.4). Unlike baseline methods that assume the same optimal composition for all data scales, our method finds optimal compositions specific to each data scale. For more experiments on CIFAR-10, we refer the reader to Appendix D.

### 5.3 Application to Fine-tuning

As powerful architectures has been introduced and computation power has improved, larger models and datasets has become increasingly prevalent in training visual and natural language tasks. However, retraining these large pre-trained foundation models can be cost-prohibitive, which leads to widespread adoption for fine-tuning these models. However, these large pre-trained foundation models are expensive to retrain but are popular for fine-tuning for more customized tasks. In our case, we adopt a pre-trained Faster R-CNN model trained on COCO dataset. Our task is to fine-tune the model on the autonomous driving dataset for object detection, BDD100K [15]. We assume each data source to specialize in taking pictures at a specific time of day, i.e. daytime, night, or dawn/dusk pictures. Similarly to the previous task, we select optimal data source composition and project the mean average precision (**mAP**) onto larger data scales. In Figure 1(b), we observe fine-tuning accuracy projection onto eight larger data scales from 1000 samples and observe that our predictions do not deviate from the actual accuracy by more than 0.4, which indicates `projektor`'s extended capability of performance projection for fine-tuning.

## 6 Discussion and Outlook

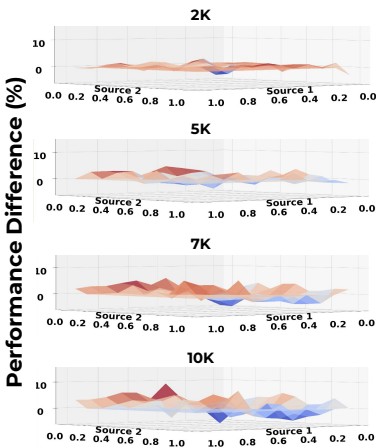

Figure 8: A direct visualization for the landscape of prediction errors in model performance for all data compositions as the scale of data to be predicted grows. We consider the setting of 3 data sources with unlabeled samples from CIFAR-10. We construct the proposed performance predictors within 1k samples that are considered accessible to the practitioner. We then predict model performance for all data compositions and on larger data scales (2-10k samples). X-Y axes represent the proportion of data from data sources #1 and #2 (consequentially, the proportion of data from data source #3 will be 100%-X-Y) and the prediction errors are visualized in Z-axis (horizontal plane represents zero error, red spikes blue dips represent deviations from over-/under-prediction). As the data scale gets larger, the performance predicted from the initial 1k samples becomes slightly less accurate, but prediction error remains mostly within 5% even at the scale of 10k data (10 times larger than the pilot data), retaining its effectiveness and functionality.

This paper presents a novel framework to conduct data selection from partially revealed sources, addressing the practical challenge in emerging data market scenarios. In particular, in contrast to existing work that tries to directly fit non-informative parametric surrogates on the limited available samples to predict model performance at different data sizes and compositions of data sources, which suffers from pronounced computational burdens and often unsatisfactory results, our key technical contribution is an OT-based performance scaling method. The *take-away* from our empirical study is that despite being extensively adopted in the past, fitting non-informative parametric surrogates for predicting performance scaling is actually suboptimal–computationally inefficient, often impractical, and less accurate; utilizing data distance in the performance prediction provides immediate benefits and presents a better pathway to construct the predictors.

Contributing a new perspective with performance and efficiency improvements, this work still has some **Limitations** and opens up many new investigation venues, such as lifting the requirement on validation data, accounting for malicious data owners, and extending to data sources that are misaligned in feature space. Additional discussions and **Broader Impacts** are provided in Appendix.E.

## Acknowledgments and Disclosure of Funding

RJ and the ReDS lab acknowledge support through grants from the Amazon-Virginia Tech Initiative for Efficient and Robust Machine Learning, the National Science Foundation under Grant No. IIS-2312794, NSF IIS-2313130, NSF OAC-2239622, and the Commonwealth Cyber Initiative.

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

# Appendices

# Appendix A `projektor` : Framework and Algorithms

## A.1 `projektor` Pipeline

**I. Training Data Preparation.** For data distance-based performance prediction, the first step of the pipeline is to prepare "training data" (introduced in Section 4.2) to fit parameters of Equations 1 and 2. The data consists of different data source compositions for some given data scales $N_0$, $N_1$, where for each composition $\mathbf{p}$, we compute the OT distance of the composed dataset $\mathcal{D}(N_1, \mathbf{p})$ to the validation data and then train a model on $\mathcal{D}(N_1, \mathbf{p})$ to get the actual model performance. For simplicity, we select compositions through grid search. This step is represented in Algorithm 1 from lines 4-9 and is the first (I) step in the pipeline Figure 2.

**II. Fitting Predictor Function.** Once the training data is prepared, we proceed to fit our function in Eqs. 1 and 2. This step is shown in lines 10-11 in Algorithm 1 and is the second (II) step of the pipeline Figure 2. Then, with the performance predictors fitted for data scales $N_0$ and $N_1$, we move on to the inference stage, where we can perform 2 tasks: performance projection and data source selection.

**III. Two-Stage Performance Projection.** For performance projection, we project the performance to any data size $N$ for any mixing ratio $\mathbf{p}$ in two stages. **(1)** We predict the performance given the mixing ratio $\mathbf{p}$ at data scales $N_0$ and $N_1$. **(2)** We use Eq. 5 to project performance prediction to any data size $N$. This process is represented in line 12 of Algorithm 1 and is the third (III) step of the pipeline Figure 2.

**IV. Optimal Data Source Selection.** For optimal data source selection, we solve an optimization problem through gradient descent, which is provided in Eq. B.3.1. The gradient computation uses parameters of the fitted functions from step II and the process terminates when the mixing ratio converges. This process is represented in line 12 of Algorithm 2 and is the fourth (IV) step of the pipeline Figure 2.

---

**Algorithm 1:** `projektor` performance predictor.

---

**In** : Pilot Datasets $D_1^{pi}, D_2^{pi}, \ldots, D_m^{pi}$; Query Data Budget $N$; Query Mixing Ratio $\mathbf{p}$; 0-Data Scale Size $N_0$; 1-Data Scale Size $N_1$; Learning Algorithm $\mathcal{A}$; Performance Metric Function $\mathcal{L}(\cdot, D^{val})$; OT Distance Function $OT(\cdot, D^{val})$.

**Out** : Projected Model Performance $\rightarrow [0, 1]$.

1   `P` $\leftarrow$ Generate mixing ratios
2   `DT`$_0$, `DT`$_1$ $\leftarrow$ Initialize empty lists to store OT distances
3   `L`$_0$, `L`$_1$ $\leftarrow$ Initialize empty lists to store performance values
4   **for** *Mixing Ratio* $\mathbf{p_i}$ *in* `P` **do**
5      $S_0, S_1 = \mathcal{D}(N_0, \mathbf{p_i}), \mathcal{D}(N_1, \mathbf{p_i})$ newly composed datasets of size $N_0, N_1$
6      `DT`$_0$ $\leftarrow$ append $OT(S_0, D^{val})$ Optimal Transport distance between $S_0$ and $D^{val}$
7      `DT`$_1$ $\leftarrow$ append $OT(S_1, D^{val})$ Optimal Transport distance between $S_1$ and $D^{val}$
8      `L`$_0$ $\leftarrow$ append $\mathcal{L}(\mathcal{A}(S_0), D^{val})$ Performance of a model trained on $S_0$
9      `L`$_1$ $\leftarrow$ append $\mathcal{L}(\mathcal{A}(S_1), D^{val})$ Performance of a model trained on $S_1$
10   $\hat{\mathcal{L}}(\mathcal{A}(\mathcal{D}(N_0, \cdot)), D^{val}) \leftarrow$ Fit the function from Eq. 2 with ((`P`, `DT`$_0$), `L`$_0$)
11   $\hat{\mathcal{L}}(\mathcal{A}(\mathcal{D}(N_1, \cdot)), D^{val}) \leftarrow$ Fit the function from Eq. 2 with ((`P`, `DT`$_1$), `L`$_1$)
12   $\hat{\mathcal{L}}(\mathcal{A}(\mathcal{D}(N, \mathbf{p})); D^{val}) \leftarrow$ Project performance by substituting $\hat{\mathcal{L}}(\mathcal{A}(\mathcal{D}(N_0, \mathbf{p})), D^{val})$ and $\hat{\mathcal{L}}(\mathcal{A}(\mathcal{D}(N_1, \mathbf{p})), D^{val})$ into Eq. 5
13   return $\hat{\mathcal{L}}(\mathcal{A}(\mathcal{D}(N, \mathbf{p}); D^{val})$

---

**Algorithm 2:** Optimal data source composition $\mathbf{p}^*$ Update.

**In** : Pilot Datasets $D_1^{pi}, D_2^{pi}, \ldots, D_m^{pi}$; Query Data Budget $N$; Query Mixing Ratio $\mathbf{p}$;
0-Data Scale Size $N_0$; 1-Data Scale Size $N_1$; Trained `projektor` Models with $N_0$ and
$N_1$ Data Budgets: $f_0, f_1$; Enquired Data Budget: $N$; OT Distance Function $OT(\cdot, D^{val})$
Validation Set: $D^{val}$.

**Out :** Optimal data source composition $\mathbf{p}^*$.

1 $\mathbf{p} \leftarrow$ Initialize Random Data Source Composition
2 **while** $\mathbf{p}$ *not converged* **do**
3     $S_0, S_1 = \mathcal{D}(N_0, \mathbf{p_i}), \mathcal{D}(N_1, \mathbf{p_i})$ newly composed datasets of size $N_0, N_1$
4     `gradient` $\leftarrow$ Compute gradient wrt. $\mathbf{p}$ using Eqs. 6, 7, 8.
5     $\mathbf{p} \leftarrow$ Update composition $\mathbf{p}$ with the `gradient` update according to Eq. B.3.1
6 **return** $\mathbf{p}$

# Appendix B    Proofs and Optimization Details

## B.1    Proof for Theorem 1

**Theorem 1** (Data Composition Dependent Performance Projection **(restated)**). *Consider log-linear performance scaling relationship depending on both data size $N$ and data composition $\mathbf{p}$ given as*

$$\mathbb{E}_V[\mathcal{L}(\mathcal{A}(\mathcal{D}(N, \mathbf{p})); D^{val})] = -\alpha(\mathbf{p})\log(N) + C(\mathbf{p}) \tag{4}$$

*Assume one has completed the fitting of the performance predictor on two different scales $N_0 < N_1$, which gives $\hat{\mathcal{L}}(\mathcal{A}(\mathcal{D}(N_0, \mathbf{p})); D^{val})$ and $\hat{\mathcal{L}}(\mathcal{A}(\mathcal{D}(N_1, \mathbf{p})); D^{val})$ for all data mixtures $\mathbf{p}$. Then, the model performance $\hat{\mathcal{L}}(\mathcal{A}(\mathcal{D}(N, \mathbf{p})); D^{val})$ for any data mixture $\mathbf{p}$ at any data scale $N$ can be predicted as*

$$\hat{\mathcal{L}}(\mathcal{A}(\mathcal{D}(N, \mathbf{p}); D^{val}) = \left(\log \frac{N_1}{N_0}\right)^{-1} \left[\log \frac{N}{N_0} \hat{\mathcal{L}}(\mathcal{A}(\mathcal{D}(N_1, \mathbf{p})); D^{val}) - \log \frac{N}{N_1} \hat{\mathcal{L}}(\mathcal{A}(\mathcal{D}(N_0, \mathbf{p}); D^{val})\right] \tag{5}$$

*Proof.* From Eq. (4), for any data mixture $\mathbf{p_z}$, we have

$$\mathbb{E}_V[\mathcal{L}(\mathcal{A}(\mathcal{D}(N, \mathbf{p_z})); D^{val})] = -\alpha(\mathbf{p_z})\log(N) + C(\mathbf{p_z})$$

Then, for data scales $N_0$ and $N_1$ where one has completed fitting the performance predictors, we have

$$\mathbb{E}_V[\mathcal{L}(\mathcal{A}(\mathcal{D}(N_0, \mathbf{p_z})); D^{val})] = -\alpha(\mathbf{p_z})\log(N_0) + C(\mathbf{p_z})$$
$$\mathbb{E}_V[\mathcal{L}(\mathcal{A}(\mathcal{D}(N_1, \mathbf{p_z})); D^{val})] = -\alpha(\mathbf{p_z})\log(N_1) + C(\mathbf{p_z})$$

which gives

$$\hat{\alpha}(\mathbf{p_z}) = \frac{\hat{\mathcal{L}}(\mathcal{A}(\mathcal{D}(N_0, \mathbf{p_z})); D^{val}) - \hat{\mathcal{L}}(\mathcal{A}(\mathcal{D}(N_1, \mathbf{p_z})); D^{val})}{\log(N_1) - \log(N_0)}$$

where $\hat{\mathcal{L}}(\mathcal{A}(\mathcal{D}(N_0, \mathbf{p_z})); D^{val})$ and $\hat{\mathcal{L}}(\mathcal{A}(\mathcal{D}(N_1, \mathbf{p_z})); D^{val})$ are given by the fitted predictors.

For any data scale $N_z$, we have

$$\mathbb{E}_V[\mathcal{L}(\mathcal{A}(\mathcal{D}(N_z, \mathbf{p_z})); D^{val})] = -\alpha(\mathbf{p_z})\log(N_z) + C(\mathbf{p_z})$$

Plugging in the above equations, the performance prediction can be given by

$$\hat{\mathcal{L}}(\mathcal{A}(\mathcal{D}(N_z, \mathbf{p_z})); D^{val}) = -\hat{\alpha}(\mathbf{p_z}) \cdot [\log(N_z) - \log(N_1)] + \hat{\mathcal{L}}(\mathcal{A}(\mathcal{D}(N_1, \mathbf{p})); D^{val})$$

$$= -[\hat{\mathcal{L}}(\mathcal{A}(\mathcal{D}(N_0, \mathbf{p_z})); D^{val}) - \hat{\mathcal{L}}(\mathcal{A}(\mathcal{D}(N_1, \mathbf{p_z})); D^{val})]\frac{\log(N_z) - \log(N_1)}{\log(N_1) - \log(N_0)}$$

$$+ \hat{\mathcal{L}}(\mathcal{A}(\mathcal{D}(N_1, \mathbf{p})); D^{val})$$

$$= \left(\log \frac{N_1}{N_0}\right)^{-1} \left[\log \frac{N_z}{N_0} \hat{\mathcal{L}}(\mathcal{A}(\mathcal{D}(N_1, \mathbf{p_z})); D^{val} - \log \frac{N_z}{N_1} \hat{\mathcal{L}}(\mathcal{A}(\mathcal{D}(N_0, \mathbf{p_z}); D^{val})\right]$$

which completes the proof.

**Q.E.D.** □

## B.2  Objectives for Data Selection

Problem formulation is provided in Section 3, where the following two objectives are introduced. Given a selection budget of $N$ samples, a mixing ratio of data sources $\mathbf{p} = \{p_1, \ldots, p_m\}$ such that $\forall_i, 0 \leq p_i \leq 1$ and $\sum_{i=1}^{m} p_i = 1$, and $m$ datasets $D_1, \ldots, D_m$ to be mixed, we denote the selected dataset by $\mathcal{D}(N, \mathbf{p}) = S_1 \cup \cdots \cup S_m$, where each $S_i$ is a random subset of $D_i^{\text{all}}$ and $|S_i| = p_i N$. Using these notations, we now describe the typical acquisition goals that can be accommodated by our approach:

- (Primary) *Fixed-budget selection for maximal performance:* The collector seeks to maximize the resulting model performance by strategically choosing the mixing ratio $\mathbf{p}$ of $m$ data sources at a *pre-specified* selection budget $N_s \leq \sum_{i=1}^{m} \bar{N}_i$. The objective can be formalized as $\max_{\mathbf{p}} \mathcal{L}(\mathcal{A}(\mathcal{D}(N_s, \mathbf{p})), D^{\text{val}})$.

- (Alternative) *Flexible-budget selection for reaching performance threshold with minimal costs:* The collector seeks to attain a target model performance $u^{\text{tar}}$ by choosing *both* the mixing ratio $\mathbf{p}$ as well as the selection budget $N$. More formally, the objective can be expressed as $\min_{N,\mathbf{p}} N$ s.t. $\max_{\mathbf{p}} \mathcal{L}(\mathcal{A}(\mathcal{D}(N, \mathbf{p})), D^{\text{val}}) \geq u^{\text{tar}}$.

The primary objective, *"fixed-budget selection for maximal performance"*, is formulated as a convex optimization and we solve it via gradient-based methods. The alternative objective can be treated as a direct extension of the primary, where one solves the *"fixed-budget selection for maximal performance"* problem for different data quantities $N$ and performs a *line search* for minimal data quantity $N$ that meets the performance requirement.

## B.3  Optimization and Convexity

### B.3.1  Primary

For our primary objective *fixed-budget selection for maximal performance*, with the proposed performance predictors with projection, we solve for

$$\mathbf{p}^* = \arg\max_{\mathbf{p}} \hat{\mathcal{L}}(\mathcal{A}(\mathcal{D}(N_s, \mathbf{p})), D^{\text{val}})$$

We show this objective function is convex in data composition $\mathbf{p}$, where the proposed gradient-based method will guarantee to find its optimal solution $\mathbf{p}^*$ efficiently.

Empirically, model performance $\mathcal{L}$ always appears convex in data composition $\mathbf{p}$, which is also reported in [13]. This means the model trained on data combined from multiple sources always achieves no worse performance than the average performance of models trained separately on data from each source. Theoretically, as given in [38], the gap between training and validation performance can be tightly bounded by the OT distance between training and validation data, whereas the OT distance is always convex in data composition $\mathbf{p}$. We now show our proposed performance predictors as well as the optimization problem based on them are also convex in data composition $\mathbf{p}$.

For `projektor/CS` in Eq. (1), consider data compositions $\mathbf{p_0} \neq \mathbf{p_1}$ and a convex combination $\mathbf{p_2} = \alpha \mathbf{p_0} + (1 - \alpha)\mathbf{p_1}$ for some constant $\alpha \in (0, 1)$. Then, we have

$$\hat{\mathcal{L}}\left(\mathcal{A}(\mathcal{D}(N, \mathbf{p_2})), D^{\text{val}}\right) - \alpha\hat{\mathcal{L}}\left(\mathcal{A}(\mathcal{D}(N, \mathbf{p_0})), D^{\text{val}}\right) - (1 - \alpha)\hat{\mathcal{L}}\left(\mathcal{A}(\mathcal{D}(N, \mathbf{p_1})), D^{\text{val}}\right)$$

$$= a_1 \cdot \left[\text{OT}\left(\mathcal{D}(N, \mathbf{p_2}), D^{\text{val}}\right) - \alpha\text{OT}\left(\mathcal{D}(N, \mathbf{p_0}), D^{\text{val}}\right) - (1 - \alpha)\text{OT}\left(\mathcal{D}(N, \mathbf{p_1}), D^{\text{val}}\right)\right]$$

$$\geq 0$$

where the inequality holds because $a_1 > 0$ always holds and OT distance is always convex in data composition $\mathbf{p}$ by definition [16]. Thus, `projektor/CS` is convex in data composition $\mathbf{p}$. `projektor/PQ` is constructed as `projektor/CS` with pseudo quadratic terms and its convexity in $\mathbf{p}$ can be shown similarly.

We solve the above optimization problem iteratively with the following procedure. First, we initialize the algorithm with $\mathbf{p} = \mathbf{p^0}$ where $\mathbf{p^0}$ can be chosen arbitrarily provided that $\sum_i p_i = 1$. Then, at each step, we perform the gradient update as

$$\mathbf{p^{t+1}} \leftarrow \mathbf{p^t} + d^t \cdot \left.\frac{\partial \hat{\mathcal{L}}(\mathcal{A}(\mathcal{D}(N_s, \mathbf{p})), D^{\text{val}})}{\partial \mathbf{p}}\right|_{\mathbf{p}=\mathbf{p}^t}$$

where $d^t$ is the step size at iteration $t$ and we obtain $\mathbf{p}^* = \mathbf{p}^\mathbf{T}$ at convergence as the desired optimal solution.

### B.3.2 Alternative

Then, for flexible-budget selection for reaching performance threshold with minimal budget:

$$\min_{N,\mathbf{p}} N \text{ s.t. } \max_{\mathbf{p}} \mathcal{L}(\mathcal{A}(\mathcal{D}(N,\mathbf{p}), D^{\text{val}}) \geq u^{\text{tar}}$$

We solve it through a bi-level optimization, where the lower level is the same as the primary objective and the upper level is a *line search* for optimal data quantity $N^*$. Note that the model performance $\mathcal{L}(\mathcal{A}(\mathcal{D}(N,\mathbf{p}), D^{\text{val}})$ is *monotonically non-decreasing* and *concave* in $N$. Thus, the optimal data quantity $N^*$ can be found via a straightforward *line search*. Initialize at $N_0 = 0$ and $\mathbf{p} = \mathbf{p^0}$ where $\mathbf{p^0}$ can be chosen arbitrarily provided that $\sum_i p_i = 1$. Then, at each step, we perform the gradient update as

$$N^{t+1} \leftarrow N^t + d^t \cdot \left. \frac{\partial \hat{\mathcal{L}}(\mathcal{A}(\mathcal{D}(N,\mathbf{p^{t*}})), D^{\text{val}})}{\partial N} \right|_{N=N^t}$$

with

$$\mathbf{p^{t*}} = \arg\max_{\mathbf{p}} \hat{\mathcal{L}}(\mathcal{A}(\mathcal{D}(N_t,\mathbf{p})), D^{\text{val}})$$

where $d^t$ is the step size at iteration $t$ and $\mathbf{p}^{t*}$ is the optimal data mixture at $N_t$, respectively. Continue until $\hat{\mathcal{L}}(\mathcal{A}(\mathcal{D}(N_t,\mathbf{p^{t*}})), D^{\text{val}}) \geq u^{tar}$ is achieved and then the data acquisition strategy $\mathcal{D}(N_t,\mathbf{p^{t*}})$ is accepted. Note that at each step, $\mathbf{p}^t$ is initialized from $\mathbf{p}^{(t-1)*}$ and the optimization for $\mathbf{p}^{t*}$ is completed fairly easily within a few steps.

### B.4 Gradients Calculation, Stepsize Selection, and Convergence

For the primary problem optimizing over $\mathbf{p}(N)$ where $N$ is the target data quantity, the gradients can be calculated as

$$\frac{\partial \hat{\mathcal{L}}(\mathcal{A}(\mathcal{D}(N,\mathbf{p}); D^{\text{val}})}{\partial \mathbf{p}} = \left( \log \frac{N_1}{N_0} \right)^{-1} \left[ \log \frac{N}{N_0} \frac{\partial \hat{\mathcal{L}}(\mathcal{A}(\mathcal{D}(N_1,\mathbf{p})); D^{\text{val}})}{\partial \mathbf{p}} - \log \frac{N}{N_1} \frac{\partial \hat{\mathcal{L}}(\mathcal{A}(\mathcal{D}(N_0,\mathbf{p}); D^{\text{val}})}{\partial \mathbf{p}} \right] \tag{6}$$

Optimal Transport naturally provides the gradient information of the OT distance w.r.t. the probability mass of datapoints on which it is computed in its dual solutions. [8] provides an approach that directly constructs the *calibrated gradient* from the output of the OT solver that informs how the OT distance changes as the probability mass of the datapoints changes, while ensuring the updated mixture $\mathbf{p}$ remains within the simplex $\sum_i p_i = 1$ at each step.

Specifically, for `projektor/CS`, recalling Eq. (1), we have

$$\frac{\partial \hat{\mathcal{L}}(\mathcal{A}(\mathcal{D}(N,\mathbf{p}); D^{\text{val}})}{\partial \mathbf{p}} = a_1 \cdot \frac{\partial \text{OT}(\mathcal{A}(\mathcal{D}(N,\mathbf{p}); D^{\text{val}})}{\partial \mathbf{p}} = \left[ \frac{\partial \text{OT}(\mathcal{A}(\mathcal{D}(N,\mathbf{p}); D^{\text{val}})}{\partial p_1} ... \frac{\partial \text{OT}(\mathcal{A}(\mathcal{D}(N,\mathbf{p}); D^{\text{val}})}{\partial p_m} \right],$$

where $p_1 + p_2 + ... p_m = 1$. Let $\{r_1^1, r_1^2 ... r_1^{n_1} ... r_i^{n_i} ... r_m^{n_m}\}$ be the samples consisting $R(N,\mathbf{p})$ where $r_i$ represents samples from data source $i$. Then, the calibrated gradient can be given as

$$\frac{\partial \text{OT}(\mathcal{A}(\mathcal{D}(N,\mathbf{p}); D^{\text{val}})}{\partial p_i} = \frac{1}{n_i} \left( \sum_{j=1}^{n_i} f_i^j - \frac{n_i}{N - n_i} \sum_{x=\{1...m\}\backslash i} \sum_{y=1}^{n_x} f_x^y \right), \tag{7}$$

where $f_i^j$ is the dual solution of OT that corresponds to $r_i^j$. The calibrated gradient ensures the updated mixture $\mathbf{p}$ remains within the simplex $\sum_i p_i = 1$ at each step. Similarly, for `projektor/PQ` in Eq. (2), the calibrated gradient is given as

$$\frac{\partial \hat{\mathcal{L}}(\mathcal{A}(\mathcal{D}(N,\mathbf{p}); D^{\text{val}})}{\partial p_i} = (b_2^i \cdot p_i^2 + b_1^i \cdot p_i + b_0) \cdot \frac{\partial \text{OT}(\mathcal{A}(\mathcal{D}(N,\mathbf{p}); D^{\text{val}})}{\partial p_i}$$
$$+ [b_2^i \cdot (2p_i) + b_1^i] \cdot \text{OT}(\mathcal{A}(\mathcal{D}(N,\mathbf{p}); D^{\text{val}}) + [c_2^i \cdot (2p_i) + c_1^i]. \tag{8}$$

Then, to perform gradient-based optimization, first, we initialize the algorithm with $\mathbf{p} = \mathbf{p^0}$ where $\mathbf{p^0}$ can be chosen arbitrarily provided at $\sum_i p_i = 1$. Then, at each step, we perform the gradient update as

$$\mathbf{p^{t+1}} \leftarrow \mathbf{p^t} + d^t \cdot \left. \frac{\partial \hat{\mathcal{L}}(\mathcal{A}(\mathcal{D}(N_s, \mathbf{p})), D^{\mathrm{val}})}{\partial \mathbf{p}} \right|_{\mathbf{p}=\mathbf{p}^t}$$

where $d^t > 0$ is the step size at iteration $t$. In practice, we choose diminishing step sizes that satisfy Robbins—Monro conditions such that $d^t < d^{t+1}$, $\sum_t d^t = \infty$, and $\sum_t (d^t)^2 < \infty$. then the series $\mathbf{p^t}$ is guaranteed to converge to the optimal solution $\mathbf{p}^*$ [44] given that objective function is convex and bounded. Gradients for `projektor/PQ` can be obtained similarly and the solution procedure is the same. The proposed method is shown to achieve remarkable performance and fast convergence, yielding satisfactory results in a swift manner.

## Appendix C   Sampling Stochasticity and Market Practices

In the data selection problem formulated in this work, we aim to optimize predicted performance with objectives given in terms of finite samples from the pilot dataset. We note that this pilot dataset is considered a random sample from the whole dataset of each data provider and is inevitably affected by the stochasticity of the sampling process. Our performance predictions $\hat{\mathcal{L}}$ are empirical estimates of the expectation for the variable based on the samples. Thus, it substantially depends on the sampling process for the estimates to be unbiased and precise. Data providers should adhere to certain guidelines when selecting the pilot datasets. The sampling process should be unbiased where each sample is selected with an equal chance. Examples include sampling with a Bernoulli process where each sample is selected with a fixed probability $p$; or permutation sampling where one selects the first $N$ samples from the random permutation.

There might be strategic providers that do not adhere to the guidelines. Multiple mechanisms are available to incentivize providers to provide true samples. Since the full dataset will be revealed after purchase, the data buyer can examine the posterior probability for the pilot dataset being sampled from the whole dataset according to the prescribed sampling protocol. The chance of the distribution of the pilot dataset having a large deviation from that of the whole dataset should be small. A threshold for hypothesis testing can be set to determine whether to accept or reject the pilot dataset as an unbiased sample from the whole dataset. If there are external supervisions (e.g., market regulators), they can conduct sequential hypothesis testing to check whether the samples provided by each seller converge to the whole dataset or comply to the prescribed sampling procedure.

## Appendix D   Experiment Details and Additional Results

### D.1   Datasets and Models

For our experiments, we use the following vision and language datasets:

| Dataset | Total Training Data | Total Test Data | Number of Classes |
|---|---|---|---|
| IMDB [45] | 25,000 | 25,000 | 2 |
| MNIST [46] | 50,000 | 10,000 | 10 |
| CIFAR-10 [47] | 50,000 | 10,000 | 10 |
| ImageNet100 [48] | 130,000 | 5,000 | 100 |
| BDD100K [15] | 70,000 | 10,000 | 10 |

Table 2: Details on datasets used in experiments.

For IMDB dataset, we trained a Long Short-Term Memory (LSTM) network model [49] for 20 epochs; for MNIST dataset, we use Support Vector Machines (SVMs) with RBF kernel. For CIFAR-10 dataset, we trained on the pre-activation Resnet with identity mappings (PreActResNet-18 [50]) for 100 epochs. We trained ImageNet-100 on ResNet-50 [51] backbone for 200 epochs with cosine annealing as the learning rate scheduler. For the autonomous driving object detection dataset, BDD100K, we fine-tune an improved pre-trained faster region proposal-CNN network (Faster-RCNN ResNet50 [52]) on COCO dataset [53] for 30 epochs.

## D.2 Details on Baseline Methods

For $N$ samples (data quantity) from $m$ data sources with a mixing ratio $\mathbf{p} = \{p_1, \ldots, p_m\}$, we consider the following baselines

**Linear:** $\hat{\mathcal{L}}(\mathcal{A}(\mathcal{D}(N, \mathbf{p}); D^{\text{val}}) := \mathbf{a}'\mathbf{p} + b \log(N) + c$, where $\mathbf{a} = \{a_0, a_1, ..., a_m\}$, $b$, and $c$ are coefficients to be fitted.

***Leave-one-out (LOO)** and **Shapley** can be considered special cases for **Linear**, where the coefficients are calculated as the marginal contribution of the data source (**LOO**) or its averaged contribution to different combinations of other data sources (**Shapley**) [18], as opposed to the least-square fitting as in **Linear**.*

**Pseudo-quadratic**: $\hat{\mathcal{L}}(\mathcal{A}(\mathcal{D}(N, \mathbf{p}); D^{\text{val}}) := \sum_{i=1}^{m}(c_2^i \cdot p_i^2 + c_1^i \cdot p_i + c_0) + b \log(N)$

**Quadratic**: $\hat{\mathcal{L}}(\mathcal{A}(\mathcal{D}(N, \mathbf{p}); D^{\text{val}}) := \sum_{i=1}^{m}(c_2^i \cdot p_i^2 + c_1^i \cdot p_i + c_0) + \sum_{i=1}^{m}\sum_{j=1}^{i}(c_3^{ij} \cdot p_i p_j) + b \log(N)$

**Rational**: $\hat{\mathcal{L}}(\mathcal{A}(\mathcal{D}(N, \mathbf{p}); D^{\text{val}}) := \sum_{i=1}^{m}\left(\sum_{j=1}^{m} c^{ij} \cdot p_j\right)^{-1} + b \log(N)$

*We fit the **Rational** baseline according to the setup detailed in [13] and to our best effort. Originally, the method is intended for predicting log loss, whereas in our case, we aim to predict model accuracy. Thus, we replaced the log loss with $\log(1 - accuracy)$ for the prediction target.*

## D.3 Performance Prediction for Unseen Data Mixtures p: Ablation Study

| Data Source | Case 1 | Case 2 | Case 3 | Case 4 | Case 5 |
|---|---|---|---|---|---|
| 1 | $\{0, 3, 6, 7\}$ | $\{1, 7\}$ | $\{1, 5, 6, 8, 9\}$ | $\{4, 6, 9\}$ | $\{0, 1, 2\}$ |
| 2 | $\{4, 5, 9\}$ | $\{0, 3, 6, 8, 9\}$ | $\{2, 7\}$ | $\{8\}$ | $\{2, 4, 6\}$ |
| 3 | $\{1, 2, 8\}$ | $\{2, 4, 5\}$ | $\{0, 3, 4\}$ | $\{0, 1, 2, 3, 5, 7\}$ | $\{3, 5, 7, 8\}$ |

Table 3: Class distributions of data sources for five different cases.

In the previous experiments, we presented results over a single data source class distribution. Here, we present a more comprehensive view of the `projektor`'s performance by running multiple times over random class distributions of data sources, according to Table 3. As shown in Table 4, for all cases, either `projektor/CS` or `projektor/PQ` achieves the highest performance, while baseline methods struggle to get close performance. Although, in many cases (1,2,3,5), Quadratic or Rational baseline obtains the lowest training data MAE, but in test performance, these methods have poor generalization scores, which indicate high overfitting to the training data. These results indicate that predicting performance from data source composition is insufficient for fitting and OT distance plays an important role in better alignment of data sources to the model accuracy.

| Method | Case 1 | Case 2 | Case 3 | Case 4 | Case 5 |
|---|---|---|---|---|---|
| `projektor/PQ` | $1.41 \cdot \mathbf{2.63}\ (\triangle\ 0.00)$ | $1.11 \cdot \mathbf{1.35}\ (\triangle\ 0.00)$ | $1.24 \cdot 1.22\ (\triangle\ 0.15)$ | $\underline{0.88} \cdot \mathbf{1.39}\ (\triangle\ 0.00)$ | $0.84 \cdot \mathbf{1.26}\ (\triangle\ 0.00)$ |
| `projektor/CS` | $1.45 \cdot 7.07\ (\triangle\ 4.44)$ | $1.12 \cdot 1.49\ (\triangle\ 0.14)$ | $1.24 \cdot \mathbf{1.07}\ (\triangle\ 0.00)$ | $0.91 \cdot 2.31\ (\triangle\ 0.92)$ | $0.97 \cdot 1.95\ (\triangle\ 0.69)$ |
| Linear | $3.31 \cdot 10.67\ (\triangle\ 8.04)$ | $5.51 \cdot 7.70\ (\triangle\ 6.35)$ | $4.35 \cdot 7.31\ (\triangle\ 6.24)$ | $5.38 \cdot 8.92\ (\triangle\ 7.53)$ | $4.44 \cdot 6.04\ (\triangle\ 4.78)$ |
| Pseudo-Quadratic | $1.35 \cdot 10.18\ (\triangle\ 7.55)$ | $1.04 \cdot 5.21\ (\triangle\ 3.86)$ | $1.82 \cdot 4.94\ (\triangle\ 3.87)$ | $1.34 \cdot 3.54\ (\triangle\ 2.15)$ | $1.82 \cdot 2.90\ (\triangle\ 1.64)$ |
| Quadratic | $\underline{1.05} \cdot 9.42\ (\triangle\ 6.79)$ | $\underline{1.02} \cdot 4.94\ (\triangle\ 3.59)$ | $\underline{0.97} \cdot 3.48\ (\triangle\ 3.41)$ | $0.93 \cdot 3.28\ (\triangle\ 1.89)$ | $0.88 \cdot 5.22\ (\triangle\ 3.96)$ |
| Rational | $1.06 \cdot 15.99\ (\triangle\ 13.36)$ | $1.09 \cdot 6.24\ (\triangle\ 4.89)$ | $1.04 \cdot 4.89\ (\triangle\ 3.82)$ | $1.04 \cdot 3.02\ (\triangle\ 1.63)$ | $\underline{0.83} \cdot 2.95\ (\triangle\ 1.69)$ |

Table 4: Multiple runs over random data source class distributions in CIFAR10. For each cell the left-column value denotes the fitting MAE value for fitting the training data and we marked in underlined green, whereas the right-column value denotes the MAE value for testing data and we marked in **bold red**.

## D.4 Extended Application to Multiple Number of Data Sources

So far, our experiments are focused solely on the cases with three data sources. To show the practicality of our method, we extend `projektor` to more challenging cases, where we have more than three data sources. Specifically, we explore the setting with 4, 5, and 6 data sources and illustrate

the results with baseline comparisons in Table 5. As we observe, our methods, both `projektor/CS` and `projektor/PQ`, achieve the best testing MAE scores as well as one of the lowest training MAE scores. While both Linear and Pseudo-Quadratic baselines are under-performing and receive poor testing MAE values. As observed in previous experiments, Quadratic baseline presents strong training data fitting but weak testing predictions. For the Rational baseline, we tried our best to fit this method, but we demonstrate that with the larger number of data sources, it is even harder to properly train this function which results in increasing errors, respectively. This experiment demonstrates the capability of our method to extend to more practical settings with multiple data sources.

| Method | 4 Sources | 5 Sources | 6 Sources |
|---|---|---|---|
| `projektor/PQ` | 1.15 · **2.83**(Δ **0.00**) | 0.99 · **3.40**(Δ **0.00**) | 1.14 · **2.53**(Δ **0.00**) |
| `projektor/CS` | 1.28 · 3.51 (Δ 0.68) | 1.23 · 4.14 (Δ 0.74) | 1.29 · 2.91 (Δ 0.38) |
| Linear | 2.12 · 4.58 (Δ 1.75) | 2.38 · 4.74 (Δ 1.34) | 1.99 · 4.03 (Δ 1.50) |
| Psuedo-Quadratic | 1.32 · 3.97 (Δ 1.14) | 1.43 · 5.24 (Δ 1.84) | 1.34 · 3.91 (Δ 1.38) |
| Quadratic | 1.06 · 3.93 (Δ 1.10) | 0.97 · 8.70 (Δ 5.30) | 1.02 · 4.67 (Δ 2.14) |
| Rational | 52.72 · 57.93 (Δ 55.1) | 611.5 · 501 (Δ 497.6) | 1034 · 1155 (Δ 1152) |

Table 5: Performance on various number of data sources (4,5,6) in CIFAR-10. For each cell the left-column value denotes the fitting MAE value for fitting the training data and we marked in underlined green, whereas the right-column value denotes the MAE value for testing data and we marked in **bold red**.

## D.5  Optimal Data Source Composition

Here, we showcase the optimal data source selection on CIFAR-10 dataset. Similarly as in Section 5.2 given a pilot dataset of size $1.5K$ from each of the three data sources, we would like to find the mixing ratio that can maximize the model performance when trained on $10K$ dataset. As illustrated in Fig. 9, `projektor/PQ` and `projektor/CS` select mixing ratios that achieve the highest model performance, gaining $4\%$ and $2\%$ in performance improvement over the best baseline method. Our methods select nontrivial mixing ratios which also outperform the uniform mixing ratio by $5\%$ and $3\%$, respectively. Furthermore, we observe in Fig. 10 that our methods also perform well in performance prediction into larger data scales. Specifically `projektor/PQ` and `projektor/CS` predictions are within $2.2\%$ discrepancy from the actual model performance for $10K$ dataset, while the best baseline method prediction has over $5\%$ error. To demonstrate `projektor` capability, we additionally present results on the case where data sources contain some mislabeled data.

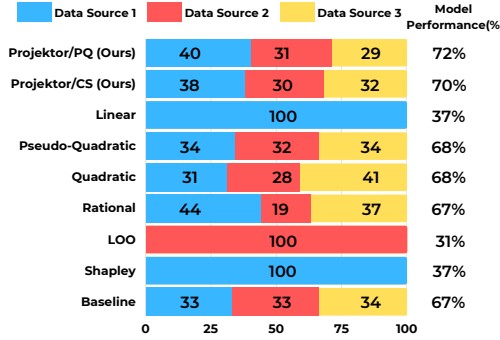

Figure 9: Optimal data source composition selection for 10K CIFAR-10 from 1.5K samples and the actual model performance.

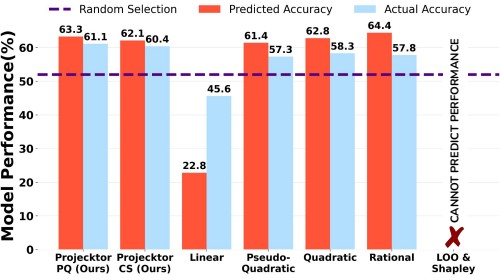

Figure 10: Performance projection of selected *optimal* mixture ratios in (Fig. 9) onto 10K CIFAR-10 from 1.5K samples. Comparison with the actual model performance.

In this case, we assume data sources have noisy labels. In particular, data sources have $20\%$, $15\%$, $25\%$ of mislabeled data, respectively. In Fig. 11, surprisingly, we observe that even though the first data source has $20\%$ of mislabeled data, choosing more of that source improves the model performance. A possible explanation is that the first source dataset contains classes that are important for learning

and the second source has less important for model performance or requires less to be learned (especially since it has a lower mislabeled rate). Moreover, we notice that `projektor`'s mixing ratio can improve the model performance by over $3\%$ from the performance of the best baseline. Results from Fig. 12 indicate that `projektor`'s performance projection onto $10K$ dataset has a prediction error within $1.7\%$, while the best baseline has an error of over $2.2\%$. To sum up, we have shown possibilities of `projektor` in mixing ratio selection and performance prediction. While baseline methods have the same mixing ratio over any data scales, our method chooses different optimal mixing ratios for different data scales, which has shown some advantages in improving model performance.

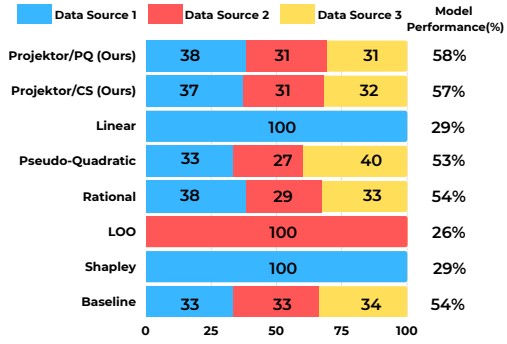

Figure 11: Optimal data source composition selection for 10K CIFAR-10 from 1.5K samples with mislabeled data sources, and the actual model performance.

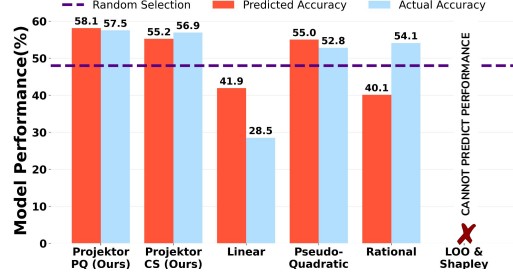

Figure 12: Performance projection of selected *optimal* mixture ratios in (Fig. 11) onto 10K CIFAR0-10 from 1.5K samples. Comparison with the actual model performance.

# Appendix E Additional Discussion and Broader Impacts

**Limitations.** Despite contributing a new perspective with performance and efficiency improvements, this work still has some limitations and opens up many new investigation venues: **(1)** How to quantify the influence or further lift the dependence on validation data? While a validation set representative of the downstream learning task is a common assumption in the ML literature, it may or may not be available during data exchange and its quality may vary. **(2)** Our design could be vulnerable to estimation errors in the scaling law for data sizes, which could lead to magnified prediction errors on larger scales and affect data acquisition decisions [54]. Especially, noises are inevitable due to the performance stochasticity of ML models as well as the sampling process to generate the pilot dataset. **(3)** Our current framework does not take into consideration of broader tasks that aim for goals beyond accuracy, e.g., fairness, variable data costs, as well as broader acquisition scenarios where data sources have misaligned feature space. Incorporating other objectives and extending to heterogeneous data sources is an exciting direction. **(4)** Our setup considers *honest* data providers and the requested samples are faithfully sampled from the actual data sources, leaving an in-depth study of potential security risks, such as malicious data manipulation [28] to future work.

**Broader Impacts.** This work will have significant impacts beyond advancing the research on data selection and data markets. The techniques developed in this work can be applied to a variety of other subfields of ML related to data acquisition, data valuation, interpretability, robustness, etc. The results of this paper will facilitate the automation of data selection and quality management in machine learning, which in turn, accelerates research and improves services based on ML. Data exchanges and data markets are also at the heart of the global data economy. The advancements in practical data exchanges in this work will substantially benefit the development of data markets and promote data sharing, contributing to the business and economy as well as society as a whole.

