# OpenReview forum: "Performance Scaling via Optimal Transport: Enabling Data Selection from Partially Revealed Sources"
_NeurIPS.cc/2023/Conference — NeurIPS 2023 poster_

### Official Review · Reviewer_WiXX · 2023-07-04

**Soundness:** 3 good
**Presentation:** 3 good
**Contribution:** 3 good
**Rating:** 6
**Confidence:** 3

**Summary:**

In this paper, the authors address the problem of data selection, acknowledging that complete data availability is often not possible, and data can only be obtained from specific providers. They recognize that different data sources may have varying impacts on the performance of the model, emphasizing the importance of optimizing the data selection strategy to enhance model training. To achieve this goal, the authors leverage optimal transport, validation data, the target model, and scaling law to allocate the selection budget effectively across different data sources. The experimental results presented in the paper validate the effectiveness of the proposed method.

Pros:

This paper is well-written, presenting a clear and logical flow of ideas, along with a comprehensive description of the method employed. The setting of the study is intriguing, considering the current landscape where data providers offer limited options for data access. Exploring the optimal combination within a constrained budget is a valuable contribution to both academia and industry research.

Cons:

1. While the chosen setting is interesting, it may be somewhat impractical since scenarios where different data providers offer data for the same task with identical distribution are uncommon. The authors implicitly assume that data from all providers share the same distribution, which might weaken the persuasiveness of the results. It would be beneficial if the authors could present a real-world scenario to support their assumptions.

2. The assumption that the selection model has access to validation data with the same distribution as the testing data might not be practical in many cases. Often, the real testing data and its distribution are unknown, making this assumption less realistic.

3. Some of the figures presented in the paper are difficult to comprehend. For instance, Figure 5 lacks clarity in terms of color interpretation, as well as the meaning of the x and y axes.

Overall, this paper offers a solid contribution to the field of data selection. Despite the presence of certain strong assumptions, the insights provided are valuable for enhancing the data selection process.

**Strengths:**

This paper is well-written, presenting a clear and logical flow of ideas, along with a comprehensive description of the method employed. The setting of the study is intriguing, considering the current landscape where data providers offer limited options for data access. Exploring the optimal combination within a constrained budget is a valuable contribution to both academia and industry research.

**Weaknesses:**

1. While the chosen setting is interesting, it may be somewhat impractical since scenarios where different data providers offer data for the same task with identical distribution are uncommon. The authors implicitly assume that data from all providers share the same distribution, which might weaken the persuasiveness of the results. It would be beneficial if the authors could present a real-world scenario to support their assumptions.

2. The assumption that the selection model has access to validation data with the same distribution as the testing data might not be practical in many cases. Often, the real testing data and its distribution are unknown, making this assumption less realistic.

3. Some of the figures presented in the paper are difficult to comprehend. For instance, Figure 5 lacks clarity in terms of color interpretation, as well as the meaning of the x and y axes.



**Questions:**

None

**Limitations:**

The assumptions are a bit strong and may degenerate the usefulness in practical setting.

---

> ### Author Rebuttal · Authors · 2023-08-09
>
> > *"impractical since scenarios where different data providers offer data for the same task with identical distribution are uncommon... implicitly assume that data from all providers share the same distribution..."* (*Weaknesses: 1*)
>
> **Re:** We appreciate the valuable feedback on the presentation of this manuscript. **We do not make assumptions about data being identically distributed. **
>
> **We rely on the fact that the distributions are different so that the discrepancy measures (such as Optimal Transport) can be leveraged to provide information from data in the construction of performance predictors.** This is also under the technical challenge of scaling laws, where the scaling ratios vary for data of different distributions. We then tackle it with the organic integration of the OT-based performance predictor to construct a self-adaptive pipeline, automatically fitting scaling parameters for the data distribution based on the prediction tools.
>
> In this work, we refined our scope to data of the same type (uni-modal)–for example, images are of the same format (i.e., resolution or the number of color channels), though, we note this framework is capable of extending to multi-modal data (e.g., multi-modal data by aligning the distribution on the joint embedding space of image and text). Our work provides the groundwork for future papers that might explore modality-misaligned data sources.
>
> We apologize for the potential unclarity and confusion it may have caused. **We would thoroughly proofread our manuscript to improve readability and prevent potential misperceptions.**
>
> ***
>
> > *"The assumption that the selection model has access to validation data with the same distribution as the testing data might not be practical in many cases.... the real testing data and its distribution are unknown, making this assumption less realistic."* (*Weaknesses: 2*)
>
> **Re:** Thanks for the interesting comment and we would be happy to discuss more about it.
>
> In general practice, data acquisition pipelines typically randomly partition the validation data into non-overlapping parts and use them for testing and validation purposes, respectively. In such cases, validation data and test data are i.i.d. **We also followed this scheme in this work.**
>
> **To implement the tools proposed in this method, the practitioner would need to first curate a validation dataset that represents the target tasks they expect the collected data to perform well on. This dataset does not need to be large, but is often considered balanced that can well represent all the target objectives.** This is generally perceived as plausible for industrial practitioners.
>
> For example, in state-of-the-art automatic speech recognition (ASR) tasks, a number of datasets are available consisting of many audio sources (e.g., accents, recording quality, environmental conditions), **the practitioners are able to curate a validation dataset representing their target user groups and scenarios** (e.g., North American market, home environment) that they hope their applications to perform best. Due to the large scale of samples, training the model each time is resource-intensive. Finding an ideal combination of data for training on the target tasks can be prohibitively expensive if conducted with manual searches. **The tools proposed in this work would provide a complete landscape of target model performance for any data composition on any scale (data quantity), helping inform practitioners in deciding what type of data to acquire and by how much so that the objectives are best met.** ***[We are adding this as a set of NEW experiments with additional results provided in the summarization rebuttal to all reviewers.]***
>
> Besides, in relevant research of data valuation *(ref [3-8] in the manuscript)* or coreset selection *(ref [2][20][21] in the manuscript)* (cited and discussed in this work), the availability of such validation datasets is also assumed. **The assumptions made in this work do not exceed those of existing research and well align with practical needs in industrial applications.**
>
> ***
>
> > *"Some of the figures presented in the paper are difficult to comprehend. For instance, Figure 5 lacks clarity in terms of color interpretation, as well as the meaning of the x and y axes."* (*Weaknesses: 3*)
>
> **Re**:  Figure 5 is a qualitative figure that is provided to facilitate the reader’s interpretation of how this framework works. We apologize if it ends up the opposite and adds to the confusion.
> ***[We provide a full response to this question in a separate comment to all reviewers. (R4)]***
>
> We will add more explanations to avoid possible misunderstandings.

---

> > ### Comment · Reviewer_WiXX · 2023-08-22
> >
> > I appreciate the authors' response. They've mostly addressed my concerns. However, when considering practical importance, relying solely on the fact that other studies make similar assumptions isn't enough to prove their point. Therefore, I'm maintaining my current score.

---

> ### Author Response · Authors · 2023-08-20
> **Rebuttal period ending–we anticipate your feedback!**
>
> Dear Reviewer WiXX,
>
> As the rebuttal/author discussion period is closing, we sincerely look forward to your feedback. The authors are deeply appreciative of your valuable time and efforts spent reviewing this paper and helping us improve it.
>
> It would be very much appreciated if you could once again help review our responses and additional results and let us know if these address or partially address your concerns and if our explanations are heading in the right direction.
>
> Please also let us know if there are any further questions or comments about this paper. We strive to consistently improve the paper and it would be our pleasure to have your precious feedback!
>
> Kind Regards,
> Authors of Submission2041

---

### Official Review · Reviewer_d24K · 2023-07-04

**Soundness:** 3 good
**Presentation:** 2 fair
**Contribution:** 2 fair
**Rating:** 5
**Confidence:** 3

**Summary:**

Developing ML systems typically requires collecting data from multiple sources. A natural question is how much data to collect from each source. This paper proposes a two-stage estimator that (1) estimates the relationship between data set proportions vs validation loss using Optimal Transport and then (2) optimizes the proportion of each source to emphasize in a dataset given a fixed budget. They validate aspects of the framework on several settings to demonstrate improved performance from good data selection decisions.

**Strengths:**

The topic is timely and of growing importance. The optimal transport perspective to data selection is novel and interesting.

**Weaknesses:**

1.	The paper makes some confusing and potentially incorrect statements regarding the related literature, particularly with power laws. For example,
   - Abstract: “these scaling functions are black-box …”. Neural scaling laws are very interpretive and give clear explanations on how data set size relates with model performance. More so, I do not see how the proposed work is less black-box than a neural scaling law.
   - The general learning curve assumed in this work is a logarithmic model of “performance $= -\alpha \log N + C$ where $\alpha, C$ are parameters to learn. While logarithmic learning curves have been used to model performance in the past, the current literature typically considers power laws. Moreover, the cited work in this part of the paper, [1], specifically discusses power laws and how they may be more effective than log learning curves (e.g., Fig 23 in [1]). While this paper is free to use a logarithmic model for ease of implementation or other design choices, the surrounding discussion is unclear about the motivation of the log learning curve as well as the implications of using log curves vs power laws in this setup.
2.	The problem of understanding how much data to collect given a fixed budget or given a performance threshold from multiple sources has been studied in several recent prior works, for example [2, 3]. It would be important to differentiate this work from this prior literature (e.g., the OT framework).
3.	The baselines considered in the numerical experiments are mostly simplified versions of the proposed method. Pertaining to the above 2 points, it would be important to validate the design decisions against more relevant related methods, for example, the data selection methods in [2, 3], or at the very least, a meaningful comparison against power law-based performance estimation strategies.
4.	While the OT-based estimator in eq. (1) is interesting, it is not well-motivated in this paper. I appreciated the interpretation of $a_1$ and hope to read an interpretation of $a_0$ as well. Furthermore, while bounds on validation loss w.r.t. OT have been studied, how tight are these bounds? This is particularly important because OT is used as an estimator rather than a bound, suggesting either (i) that OT can very tightly bound validation loss or (ii) that the constant term $a_0$ is carrying a lot of the water in this bound. This is particularly interesting because $a_0$ is assumed to be independent of data set size.
5.	If I understand correctly, in the numerical experiments, the initial pilot training data set is fixed to 55% of the full data set size. This generally makes the learning curve estimation problem significantly easier as we are well in the power law stage of the learning curve at this point. Moreover, this experiment setup weakens the motivation of having access to only a small pilot data subset. It would more interesting to consider meaningfully smaller data fractions.


[1] Jared Kaplan, Sam McCandlish, Tom Henighan, Tom B Brown, Benjamin Chess, Rewon Child, Scott Gray, Alec Radford, Jeffrey Wu, and Dario Amodei. Scaling laws for neural language models. arXiv preprint arXiv:2001.08361, 2020.

[2] Tejero, Javier Gamazo, Martin S. Zinkernagel, Sebastian Wolf, Raphael Sznitman, and Pablo Márquez-Neila. "Full or Weak annotations? An adaptive strategy for budget-constrained annotation campaigns." In Proceedings of the IEEE/CVF Conference on Computer Vision and Pattern Recognition, pp. 11381-11391. 2023.

[3] Mahmood, Rafid, James Lucas, Jose M. Alvarez, Sanja Fidler, and Marc Law. "Optimizing data collection for machine learning." Advances in Neural Information Processing Systems 35 (2022): 29915-29928.



**Questions:**

Please see the above weaknesses. Most importantly:

1. How does the log learning curve estimation framework in the proposed setup compare vs power law-based estimation strategies? Can we use a power law estimator rather than a log one? How would that affect the empirical performance?
2. Can you give better interpretation of the OT-based estimator? For example, it would be nice to read out an OT-based bound on validation loss and then show how the proposed estimator’s components capture the different elements of the bound.
3. How does the methodology compare against the related prior literature on estimating and optimizing data set sizes?




**Limitations:**

Limitations are reasonably discussed in the Appendix.

---

> ### Author Rebuttal · Authors · 2023-08-09
>
> > *"This paper proposes a two-stage estimator that (1) estimates the relationship between data set proportions vs validation loss using Optimal Transport and then (2) optimizes the proportion..."* (*Summary*)
>
> **Re:** The authors would like to thank the reviewer for providing detailed and in-depth feedback and comments on the manuscript. **We would like to first clarify a potential misperception and the concerning confusion it may have caused.**
>
> This work proposes a two-stage scheme for **performance prediction (Stage I) and projection onto large scales (Stage II)**, referred to as **"performance scaling with data composition"** and **"performance scaling with data quantity"** in the manuscript, **respectively. The use of term "performance scaling" may be somewhat abused that causes unnecessary ambiguities in different contexts.** ***[Due to its critical importance and the space limit here, we are providing the full response in a separate comment below (C1).]***
>
> ***
>
> > *"...some confusing and potentially incorrect statements regarding the related literature, particularly with power laws. For example, Abstract: “these scaling functions are black-box …” "* (*Weaknesses: 1.1*)
>
> **Re:** The authors acknowledge the responsibility for the choice of phrases that may have caused unnecessary ambiguities under different contexts and apologize for the confusion it may have caused. **As discussed above, we may have abused the term "performance scaling" and used it for different meanings in contexts outside of scaling laws (w.r.t. data quantity/data scales).**
>
> **We agree with the reviewer that these scaling laws are intuitive and interpretable, and "black-box functions" are references to the use of non-informative surrogates in representations of the relationship between data composition from different sources and the resulting model performance ("performance scaling with data composition").** ***[Due to its critical importance and the space limit here, we are providing the full response in a separate comment below (C2).]***
>
> ***
>
> > *"... the current literature typically considers power laws... [1]... specifically discusses power laws and how they may be more effective than log learning curves..."* (*Weaknesses: 1.2*)
>
> **Re:** Thanks for the excellent comment. We are appreciative of your insights and pointing out our negligence in elaboration of equations. **In short, we are using power laws rather than log functions. It looks like log-linear relationships because we brought everything to log space for better numerical properties**–we took log operations on both sides of the equation and the power law in the original space became log-linear in the log space. ***[Due to its critical importance and the space limit here, we are providing the full response in a separate comment below (C3).]***
>
> ***
>
> > *"...it would be important to validate the design decisions against more relevant related methods, for example, the data selection methods in [2, 3], or … against power law-based performance estimation strategies."* (*Weaknesses: 2, 3*)
>
> **Re:** We appreciate the reviewer for providing additional references and we would be glad to discuss them. At the time of submission, we were aware of [3] and cited a related paper [4] while [2] was published after our submission. **These papers are indeed interesting and provide inspiration for the line of research on data selection/data acquisition, though, we would like to point out that their scope and target problems are different from ours and the conceptual and technical contributions are non-overlapping.** ***[Due to its critical importance and the space limit here, we are providing the full response in a separate comment below (C4).]***
>
>
> ***
>
> > *"... while bounds on validation loss w.r.t. OT have been studied, how tight are these bounds? ... suggesting either (i) that OT can very tightly bound validation loss or (ii) that the constant term is carrying a lot of the water in this bound."* (*Weaknesses: 4*)
>
> **Re:** This is also an excellent question. We would appreciate the chance to thoroughly explain about it. ***[Due to its critical importance and the space limit here, we are providing the full response in a separate comment below (C5).]***
>
>
> ***
>
> > *“...the initial pilot training data set is fixed to 55% of the full data set size… makes the learning curve estimation problem significantly easier …weakens the motivation of having access to only a small pilot data subset… consider meaningfully smaller data fractions.”* (*Weaknesses: 5*)
>
> **Re: We apologize for the misunderstanding. 55% is for the part of work on predicting model performance on data composited from multiple data sources with arbitrary combinations (i.e., the proportion of data from each source), which is irrelevant to data quantity.** When fitting this relationship, we limit the proportion of data from each data source to no more than 55% such that we can test the **extrapolation performance** of the fitted predictors–to predict the performance of data compositions where the proportion of data from a data source exceeds 55%. Since the actual performance of these data compositions is not covered in fitting the predictors, we will be able to examine whether these fitted relationships suffer from overfitting that leads to excessive deviations in extrapolation tasks. **This is the first part of the work where we construct predictors on the pilot data with limited samples. We set the size of pilot data to 10%-20% of the larger data scales (data quantity).**
>
> **In the later part of the work, we extend the predictions to project them onto larger data scales (data quantity) where the target data scales are 5- to 10-fold of the pilot data.** E.g, for CIFAR-10, we assume access to 1k samples and project the performance to larger scales of 2k-10k. For ImageNet100, given 10k samples, we project performance to up to 50k samples. For the BDD100K dataset, we project from 1K samples to up to 5K samples.

---

> > ### Author Response · Authors · 2023-08-10
> > **C1: “performance scaling” and “scaling laws”**
> >
> > > *"This paper proposes a two-stage estimator that (1) estimates the relationship between data set proportions vs validation loss using Optimal Transport and then (2) optimizes the proportion..."* (*Summary*)
> >
> > **Re:** The authors would like to thank the reviewer for providing detailed and in-depth feedback and comments on the manuscript. **We would like to first clarify a potential misperception and the concerning confusion it may have caused.**
> >
> > This work proposes a two-stage scheme for **performance prediction (Stage I) and projection onto large scales (Stage II)**, referred to as **"performance scaling with data composition"** and **"performance scaling with data quantity"** in the manuscript, **respectively.**
> >
> > **The first stage is to obtain a functional relationship between data composition from different sources and the resulting model performance, which is achieved leveraging Optimal Transport data distances.** Then the function can be used to construct a predictor for model performance on arbitrary data composition. **This stage is conducted on the small pilot dataset, and is referred to as "performance scaling with data composition" in the manuscript.** The conceptual contribution is to incorporate data distances (via Optimal Transport) into the representation of the functional relationship. Compared to previous works that rely on fitting non-informative surrogates, the construction of the OT-informed predictors is orders of magnitude faster (efficient and scalable) and the resulting prediction tools fundamentally prevent large deviations from overfitting high-order nonlinear surrogates (robust and reliable)
> >
> > **Then, the second stage is to project the prediction of model performance to target data scales (data quantity) that are much larger than that of the pilot data.** The technical contribution is to organically incorporate the performance predictor with the scaling laws in a parameter-free projection. **This stage is referred to as "performance scaling with data quantity" in the manuscript.** In previous works, a prominent challenge in scaling laws is that the model performance for different data would scale at different rates thus constructing a universal predictor for data from arbitrary combinations of sources using fixed scaling laws could lead to unsatisfactory results. The novel integrated performance projection proposed in this work allows for predicting model performance on any data composition and data quantity in a self-adaptive manner, automatically fitting scaling parameters for the data based on the prediction tools.
> >
> > So far, main technical challenges for the conceptual problem have been solved. The proposed pipeline with the two-stage scheme would provide a complete landscape of target model performance for any data composition on any scale (data quantity), helping inform practitioners in decision making. **Additionally, we showcase how this can help find the precise optimal operating point, which is formulated as an optimization problem** based on the predicted performance and solved via efficient gradient-based methods. This framework is highly extendable and allows the practitioner to make tradeoffs on what type of data to acquire and by how much so that the objectives are best met. In empirical studies, improved performance is also demonstrated, outperforming comparable baselines in many important aspects.
> >
> > The authors acknowledge the responsibility for the presentation of this work and appreciate having the chance to revise the manuscript accordingly for better readability. **Especially, the use of term "performance scaling" may be somewhat abused that causes unnecessary ambiguities in different contexts.** We are considering replacing the phrase “performance scaling with data composition” with “functional relationship between model performance and data composition”, and reserves the term “scaling” exclusively for the scaling laws w.r.t. data quantity (data scales). **The authors would appreciate it if the reviewer could recommend phrases that have better clarity in the context.**

---

> > ### Author Response · Authors · 2023-08-10
> > **C2: black-box scaling relationships**
> >
> > > *"...some confusing and potentially incorrect statements regarding the related literature, particularly with power laws. For example, Abstract: “these scaling functions are black-box …” "* (*Weaknesses: 1.1*)
> >
> > **Re:** The authors acknowledge the responsibility for the choice of phrases that may have caused unnecessary ambiguities under different contexts and apologize for the confusion it may have caused. **As discussed above, we may have abused the term "performance scaling" and used it for different meanings in contexts outside of scaling laws (w.r.t. data quantity/data scales).**
> >
> > In the manuscript, we refer to the functional relationship between data composition from different sources and the resulting model performance as "performance scaling with data composition" and refer to the projection of the predicted model performance onto larger data scales (data quantity) as "performance scaling with data quantity", where **the latter has been more commonly associated with the term "performance scaling".**
> >
> > **We agree with the reviewer that these scaling laws are intuitive and interpretable, and "black-box functions" are references to the use of non-informative surrogates in representations of the relationship between data composition from different sources and the resulting model performance ("performance scaling with data composition").** Non-informative surrogates–e.g., rational functions [a] predicts the performance solely based on the size of data or its composition ratios (how much from each data source) **while neglecting the information of the content of data.** High-order nonlinear functions are essentially black boxes as the **implication of their parameters becomes impossible to interpret.** For example, [a] predicts the model performance L as the following
> >
> > $L\_{\lambda}(p)=\frac{1}{\lambda\_{11}\cdot p\_1+\lambda\_{12}\cdot p\_2...+\lambda\_{1k}\cdot p\_k}+\frac{1}{\lambda\_{21}\cdot p\_1+\lambda\_{22}\cdot p\_2...+\lambda\_{2k}\cdot p\_k}...+\frac{1}{\lambda\_{k1}\cdot p\_1+\lambda\_{k2}\cdot p\_2...+\lambda\_{kk}\cdot p\_k}$
> >
> > where k is the number of data sources, $p=\\{p_1, p_2... p_k\\}$ are the proportion of data from each data source, $\lambda\_{ij}$ are a number of $k^2$ parameters of the surrogate to be fitted. **With these inherently nonlinear rational functions being added together, there is no way to associate parameters $\lambda$ to the effects of data from each source or interpret how each data source contributes to the predicted model performance.**
> >
> > Again, the authors apologize for the ambiguity and the confusion it may have caused. We appreciate the reviewer for pointing these out to us and for the effort in helping improve the presentation of this work.
> >
> > > *[a] Tatsunori Hashimoto. Model performance scaling with multiple data sources. ICML, 2021.*

---

> > ### Author Response · Authors · 2023-08-10
> > **C3: log-linear vs. power law scaling laws**
> >
> > > *"... the current literature typically considers power laws... Moreover, the cited work in this part of the paper, [1], specifically discusses power laws and how they may be more effective than log learning curves..."* (*Weaknesses: 1.2*)
> >
> > **Re:** Thanks for the excellent comment. We are appreciative of your insights and pointing out our negligence in elaboration of equations. **In short, we are using power laws rather than log functions. It looks like log-linear relationships because we brought everything to log space for better numerical properties**–we took log operations on both sides of the equation and the power law in the original space became log-linear in the log space.
> >
> > In existing studies on scaling laws (e.g., [1]), the variable of interest is the validation loss, which is often the cross-entropy loss as in language tasks. In our paper, we center on directly predicting the target model "performance". For practitioners, it is typically the accuracy for classification tasks or MSE (mean squared error) for regression tasks. **Following the line of research on data selection/data acquisition, we use the residual error as our prediction variable**, which is (100%-accuracy) for classification tasks and MSE for regression tasks. The reduction of residual error is often exponential (e.g., accuracy 90%->99% with residual error 10%->1%), **and working in the log space often gives better numerical properties.**
> >
> > Then, when implementing the scaling laws, for the **power law relationship** in the original space given as
> >
> > > $L = a N^{-\gamma}$
> >
> > **taking log operation on both sides, we have**
> >
> > > $\log L = \log (a N^{-\gamma}) = \log a - \gamma \log N$
> >
> > **where $\log a$ and $-\gamma$ are the constants $C$ and $-\alpha$ in our expressions, $L$ and $\log L$ are the residual error in the original space and log space, respectively.**
> >
> > **We admit that we were unaware of the line of research studying directly using log-linear relationships to fit the scaling laws, thus not paying adequate attention to clearly stating the expressions and their derivations to distinguish them from alternative forms of scaling laws.**
> >
> > We appreciate the reviewer for providing us with additional information on current investigations on scaling laws (Figure 23 is especially helpful to us for a better understanding of their difference). The authors apologize for the confusion and will improve the presentation.

---

> > ### Author Response · Authors · 2023-08-10
> > **C4: Comparison to existing data selection methods**
> >
> > > *"...validate the design decisions against more relevant related methods… data selection methods in [2, 3], or … against power law-based performance estimation strategies."* (*Weaknesses: 2, 3*)
> >
> > **Re:** We appreciate the reviewer for providing additional references and we would be glad to discuss them. At the time of submission, we were aware of [3] and cited a related paper [4] while [2] was published after our submission. **These papers are indeed interesting and provide inspiration for the line of research on data selection/data acquisition, though, we would like to point out that their scope and target problems are different from ours and the conceptual and technical contributions are non-overlapping.**
> >
> > **Both of these works are built for iterative data acquisition**, where the data collector aims to acquire data in an adaptive manner and gradually improve the strategy until the target amount of data is collected or other end goals are reached (e.g., time). **These frameworks aim to collect the best set of data at the end, not to directly predict the target model performance on all data compositions from the beginning.** This is a different field of research. There are tasks where such settings align with the need (such as the image annotation task [2] is built for), but are distinct from the data market problem we set our work into. **We focus on one-shot decision processes–given a small amount of pilot data (e.g., samples), we aim to directly provide the complete landscape of target model performance for any data composition on any scale (data quantity) such that the practitioner can be informed to decide the data acquisition strategy.**
> >
> > This is also true for the industrial instances that this work intends to apply to. **For example, training large models typically cannot accommodate changing training data during the process**. For example, in state-of-the-art automatic speech recognition (ASR) tasks, a number of datasets are available consisting of many audio sources (e.g., accents, recording quality, environmental conditions), the practitioners are able to curate a validation dataset representing their target user groups and scenarios (e.g., North American market, home environment) that they hope their applications to perform best. Due to the large scale of samples, training the model each time is resource-intensive. Finding an ideal combination of data for training on the target tasks can be prohibitively expensive if conducted with manual searches. The tools proposed in this work would provide a complete landscape of target model performance for any data composition on any scale (data quantity), helping inform practitioners in deciding what type of data to acquire and by how much so that the objectives are best met. ***[We are adding this as a set of NEW experiments with additional results provided in the summarization rebuttal to all reviewers (R1).]***
> >
> > **Iterative data acquisition frameworks are not designed to accurately predict target model performance from the beginning.** Both [2] and [3] use surrogate models to represent the relationship between utility function (e.g., model performance) and the amount and proportion of data acquired from each source. In these works, [2] uses Gaussian Process (GP) and [3] uses Kernel Density Estimation (KDE), **which are both non-informative surrogates**–i.e., they predict the performance solely based on the size of data or its composition ratios (how much from each data source) while neglecting the information of the content of data. **KDE is a nonparametric method and its predictions are simply interpolations smoothened by the kernel function, while GP’s predictions are based on the estimated mean performance of each data source and the covariance between each two data source pair, resembling quadratic predictors (the quadratic baselines in our work).** Both of these methods need to fit on the model performance from a substantial number of model training on different data to function properly, and in both [2] and [3], **these models rely on adaptive improvements of their accuracy during the iterative data acquisition process. On a small amount of data (such as the pilot dataset considered in our work), their initial estimates are inaccurate and not intended for making final selections.**
> >
> > **Data selection via performance prediction based on scaling laws considers *only* the data size and not its composition** (which data sources it is from or/and by how much). As far as we are aware, [5] (cited in our work) provides a benchmark result on this strategy ("datasize" baseline). **Given that its performance is subpar compared to other baselines which consider both data composition and data size, we omit it in this work.**
> >
> > > *[4] Mahmood, Rafid, et al. How much more data do i need? estimating requirements for downstream tasks. CVPR, 2022*
> >
> > > *[5] Tatsunori Hashimoto. Model performance scaling with multiple data sources. ICML, 2021*

---

> > ### Author Response · Authors · 2023-08-10
> > **C5: OT bounds and performance prediction**
> >
> > > *"... while bounds on validation loss w.r.t. OT have been studied, how tight are these bounds? ... suggesting either (i) that OT can very tightly bound validation loss or (ii) that the constant term is carrying a lot of the water in this bound."* (*Weaknesses: 4*)
> >
> > **Re:** This is also an excellent question. We would appreciate the chance to thoroughly explain it.
> >
> > Starting from the classic result on **Kantorovich-Rubinstein Duality** (KR-duality [a])
> >
> > $W (p, q) = \inf\_{\pi∈\Pi(p,q)} E_{(x,y)∼\pi} [||x − y||\_{2}]= \sup\_{||h||_L\leq k} \left[ E\_{x∼p} [h(x)]− E\_{y∼q} [h(x)] \right]$
> >
> > which gives that the 1-Wassersterin distance (Wasserstein distance is the distance defined by Optimal Transport) between two distributions p and q upper bounds the gap between the expected empirical performance of some model h trained on samples x from p and y from q, assuming the model h is k-Lipschnitz. **The bound is tight and is attained when the Lipschitz constant k is minimal everywhere for the model on the data manifold of interest. For modern machine learning problems with neural network models, training error can be reduced to near-zero and Wasserstein distance would provide a direct indicator of expected validation performance.**
> >
> > **Yet, in practice, the precise value of Lipschitz constant is hardly known in priori.** Besides, in practical problems, we do not have access to the actual representations of underlying distributions p and q and instead, we use empirical distributions from their samples as approximations to the underlying distributions. This introduces **<sample noises>** to the calculation of Wasserstein distance. Also, OT problems are solved numerically using the efficient Sinkhorn algorithm, which carries some small approximation error from entropy regularization and adds **<entropy bias>** to the computed Wasserstein distance. **Despite the bound being tight in principle, the empirical Wasserstein distance calculated from finite samples and entropy-penalized OT includes certain noises added to the true Wasserstein distance between underlying distributions. These noises are typically invariant for the same problem and only depend on the sample size.**
> >
> > Thus, we use an affine transformation to represent the relationship between empirical Wasserstein distance and model performance of interest, which is a natural and simple choice.
> >
> > The affine transformation takes 2 parameters that represent the slope and intercept, respectively. **We refer to this approach as "center-scaling"**, where we denote slope as "scaling ratio" and intercept as "centering constant", corresponding to fitting the terms in the above-described relationship. **Intuitively, the "scaling ratio" serves as an empirical estimation for the value of Lipschitz constant on the data manifold and the "centering constant" fits the invariant noises in empirical Wasserstein distance and aligns the predictor with the model performance.**
> >
> > These values are important in connecting the empirical performance of the model to the Wasserstein distance between training and validation data, but are impossible to be obtained analytically. **This work develops a new approach that demonstrates the plausibility of estimating these quantities empirically and successfully constructs predictors and develops applications based on these estimated relationships.** This work sheds light on a new path that shall provide inspiration for future work on data selection, data valuation, performance prediction, etc.
> >
> > We would compile this discussion into a section into the Appendix for the benefit of readers.
> >
> > Additionally, the technical pipeline of this work is built on the novel result of class-wise hierarchical OT distance first proposed in [b], which treats distances between labels as the OT distance between their features, where the analysis resembles the classic Kantorovich-Rubinstein Duality. For detailed derivations, please refer to [c] where comprehensive elaborations are provided in its Appendix.
> >
> > > *[a] David A Edwards. On the kantorovich–rubinstein theorem. Expositiones Mathematicae, 29(4):387–398, 2011.*
> >
> > > *[b] Alvarez-Melis, David, and Nicolo Fusi. "Geometric dataset distances via optimal transport." Advances in Neural Information Processing Systems 33 (2020): 21428-21439.*
> >
> > > *[c] Just, Hoang Anh, et al. "LAVA: Data Valuation without Pre-Specified Learning Algorithms." The Eleventh International Conference on Learning Representations. 2022.*

---

> > > ### Comment · Reviewer_d24K · 2023-08-13
> > > **Thanks for the rebuttal**
> > >
> > > Thanks for the detailed rebuttal. I agree that using “performance scaling with data composition" and "performance scaling with data quantity" simultaneously is confusing and I agree with your recommendation of using “functional relationship between model performance and data composition”, and reserving “scaling” exclusively for scaling laws.“
> > >
> > > The paper will also improve with a more clearer discussion of the related scaling law literature, including the clarification w.r.t. log relationships and power law relationships, as well as interpretiveness with regards to multiple data sources.
> > >
> > > > These frameworks aim to collect the best set of data at the end, not to directly predict the target model performance on all data compositions from the beginning. This is a different field of research.
> > >
> > > I find this differentiation unclear. To me, predicting target model performance as a function of data set size and combination and collecting the best set of data at the end are practically the same problems as they are used for the same downstream objective (i.e., deciding the best data acquisition strategy). For example with respect to the industrial applications listed in the rebuttal, it seems that [2, 3] could be used as a baseline.
> > >
> > > I thank again for the clarification and in light of the response am happy to raise my score somewhat. I encourage you to incorporate a lot of this discussion into the revised paper.

---

> > > > ### Author Response · Authors · 2023-08-15
> > > > **On distinctions in data acquisition**
> > > >
> > > > The authors express their sincere appreciation to the reviewer for devoting the time and effort in helping to develop this paper. We thank the reviewer for the willingness to reassess this manuscript despite its presentation issues. We will thoroughly revise the presentation of the paper in light of this discussion. Your valuable feedback will greatly help improve the quality of this paper as well as our own understanding of it.
> > > >
> > > > There are two major reasons behind this distinction between "iterative data acquisition" and "one-shot performance prediction"–**1. iterative schemes are hard to deploy for larger models due to their computational demand and scalability issues (such as in the ASR tasks); 2. current business models for data transactions and data markets only support one-shot purchasing decisions.** These two research lines are independent but both often use the keyword **"data acquisition"**.
> > > >
> > > > **1. Technically**, for iterative data acquisition schemes, the standard pipeline can be described as the following
> > > >
> > > > - request a small batch of data from the data sources
> > > > - use it to infer the quality of data from each source
> > > > - update the estimated information for each source
> > > > - use the updated estimates to improve the next data request decision
> > > > - …loop until the target amount of data is collected or certain objectives are met.
> > > >
> > > > These methods rely on repetitive updates of their estimates on how data from each source contributes to the objectives and adaptively improves data selection decisions. **Each time a new batch of data is collected, the model needs to be re-trained (potentially many times) to get its new performance results on the current dataset–1. It does not matter if the initial few requests are non-optimal; 2. It needs to keep re-training the model as the collected dataset grows and until it reaches the target size.**
> > > >
> > > > In the scenario we consider, the decision must be made with only a small amount of data. **Training the model on larger data sizes is prohibitively expensive and repetitive retraining is unaffordable.**
> > > >
> > > > As in the automated speech recognition tasks considered in our evaluation, fine-tuning a pre-trained model on a smaller benchmark dataset LibriSpeech (as CIFAR-10 in vision tasks) needs **hundreds of GPU hours, orders of magnitudes more** than training a vision model on CIFAR-10. **Our data selection decisions are made with samples that are only 1% of the target data size**, which still takes 2~3 hours per model evaluation. **Iterative data acquisition schemes do not need to consider their data selection performance at this small proportion of data (corresponding to their initial few selection decisions).**
> > > >
> > > > Actually, the experiments in [2] only consider datasets up to a few thousand samples and there are no comparable baselines (only fixed strategies: "randomly sample images... according to a specified and fixed proportion"). Similarly, the largest dataset considered in [3] is CIFAR-10/100 and the only comparison is with power laws (which only considers data size and not at all its compositions).
> > > >
> > > > **In this regard, our work embodies significantly more practical considerations and includes substantially broader experiments and comparisons.** Those work shall have their own target applications. But at least in the scenarios this work positions in, those methods will have prominent problems at the deployment and our proposed framework would be advantageous on multiple fronts.
> > > >
> > > > **2. In terms of research genres**, iterative data acquisition has a historical line of work dated to at least two decades ago (e.g., [a] (2002) uses active learning for annotating unlabeled data). **They usually consider collecting data from web applications or sensor channels. These types of data naturally come as streams and thus are reasonable for adaptive acquisition strategies.** Both [2, 3] are representative works for this field of research.
> > > >
> > > > On the contrary, the "one-shot performance prediction" considered in this work is rooted in **emerging applications of data transactions in data markets. The stylized problem formulation corresponds to actual problems that are yet to be solved.**
> > > >
> > > > In current data markets (some are cited and discussed in this work), there are a number of sellers providing datasets for sale with a small number of free samples. **The only option for the buyer is to buy it or not.** There are active debates in the research community on whether this is reasonable. It is likely there is more business logic behind that is yet to be understood. This is still a quickly evolving business model and we may continue to see changes.
> > > >
> > > > **This stylized problem considered in this work is based on the status quo of this novel industry. We are open to consistently adapting our framework as the practical needs evolve.**
> > > >
> > > > > *[a]. Zheng, Z., & Padmanabhan, B. (2002). On active learning for data acquisition. In 2002 IEEE International Conference on Data Mining, 2002. Proceedings. (pp. 562-569). IEEE*

---

> > > > > ### Author Response · Authors · 2023-08-20
> > > > > **Rebuttal period ending–we anticipate your feedback!**
> > > > >
> > > > > Dear Reviewer d24K,
> > > > >
> > > > > As the rebuttal/author discussion period is closing, we sincerely look forward to your feedback. Once again, the authors express their sincere appreciation for your valuable time and efforts spent reviewing this paper and helping us improve it.
> > > > >
> > > > > We are compiling the discussions during the review into the manuscript. To better improve its clarity, it would be also very much appreciated if you could help review our response and let us know if it addresses or partially addresses your concerns on the differentiation with related works and if our explanations are heading in the right direction.
> > > > >
> > > > > Please also let us know if there are any further questions or comments about this paper. We strive to consistently improve the paper and it would be our pleasure to have your precious feedback!
> > > > >
> > > > > Kind Regards,
> > > > > Authors of Submission2041

---

### Official Review · Reviewer_WNMz · 2023-07-08

**Soundness:** 3 good
**Presentation:** 4 excellent
**Contribution:** 3 good
**Rating:** 6
**Confidence:** 4

**Summary:**

This paper considers the problem of predicting model performance (and subsequent data selection) under a partially revealed setting. The two challenges are estimating the right proportions as well as extrapolating to dataset scales beyond the observed scales. The paper proposes a two-stage approach called projektor: in the first stage, it predicts model performance as a simple function
(either an affine transformation of a quadratic) of the optimal transport distance between target distribution D_val and the input distribution. In the second stage, the performance is extrapolated using ideas and functional forms from neural scaling laws.
The paper also proposes a gradient-based method for data selection using the model performance estimator.

**Strengths:**

- Novel technical solution to a well motivated problem. The proposed approach is well-motivated, interesting, and well presented.
- Well written overall: motivates problem well (section 1), contextualize well within related work (section 2), clear set up (section 3),


**Weaknesses:**

- The experiments are a bit disappointing in terms of their practicality.
As far as I can tell, the data sources are just different subsets of distributions of fixed datasets (potentially unlabeled or mislabeled). Given the recent developments in the field (training language/diffusion models on large unfiltered data sources), I would have liked to see more experiments on more realistic/noisy data sources. Even for ImageNet, a better data source would be raw images from different online sources (e.g., Flickr),. Hence, it's a bit hard to judge the practical utility of the proposed approach (beyond the settings considered in the paper, which I am not sure are settings in practice where people need much better data selection strategies).
- Section 5 (evaluation) is a bit hard to follow in terms of setup: what exactly are the datasets used, etc.


**Questions:**

- In stage two, is the intuition that by fitting the two different scales N0 and N1 in stage 1, you eliminate the need to explicitly estimate the scaling parameters (alpha and C) in the scalings laws?
- How are OT distances computed for images? Is it done in some feature space? which one?

**Limitations:**

Fine.

---

> ### Author Rebuttal · Authors · 2023-08-09
>
> > *"...experiments are a bit disappointing in terms of their practicality ...would have liked to see more experiments on more realistic/noisy data sources. "* (*Weaknesses: 1*)
>
> **Re:** We appreciate the review for the crisp understanding of the conceptual narrative of this work, and would like to take this chance to discuss a bit more on the considerations in designing empirical studies for this work.
>
> **In short, precisely as the reviewer pointed out, this data selection pipeline developed in this work could potentially apply to a variety of downstream tasks in the real-world.** ***[We provide a clear and complete presentation of experiments conducted in this work in the summarization rebuttal to all reviewers (R1).]***
>
> Besides, we are in active collaboration with industrial researchers on a variety of problems. **Besides, in collaboration with industrial partners, we succeeded in deploying the proposed data selection pipeline in state-of-the-art automatic speech recognition (ASR) use cases.** ***[We are adding this as a set of NEW experiments with additional results provided in the summarization rebuttal to all reviewers (R2).]*** In ASR tasks, a number of datasets are available consisting of many audio sources (e.g., accents, recording quality, environmental conditions), the practitioners are able to curate a validation dataset representing their target user groups and scenarios (e.g., North American market, home environment) that they hope their applications to perform best. Due to the large scale of samples, training the model each time is resource-intensive. **Finding an ideal combination of data for training on the target tasks can be prohibitively expensive if conducted with manual searches.** The tools proposed in this work provide a complete landscape of target model performance for any data composition on any scale (data quantity), helping inform practitioners in deciding what type of data to acquire and by how much so that the objectives are best met.
>
> **Together with the experiments presented in the manuscript, we demonstrate a diversity of tasks with real-world instances, practical considerations, and realistic settings, showcasing the versatile capabilities of the proposed framework as well as the significance of the potential impact on both industrial applications and academic research.**
>
> ***
>
> > *"Section 5 (evaluation) is a bit hard to follow in terms of setup: what exactly are the datasets used, etc."* (*Weaknesses: 2*)
>
> **Re:**  We apologize for lack of clarity in experimental details and will address them properly here. ***[We provide a full response to this question in the summarization rebuttal to all reviewers (R1).]***
>
> ***
>
> > *"In stage two, is the intuition that by fitting the two different scales N0 and N1 in stage 1, you eliminate the need to explicitly estimate the scaling parameters (alpha and C) in the scalings laws?"* (*Questions: 1*)
>
> **Re: Exactly!** Function fitting is replaced with a direct mapping. Simple, natural, and quite effective in solving the practical challenge.
>
> ***
>
> > *"How are OT distances computed for images? Is it done in some feature space? which one?"* (*Questions: 2*)
>
> **Re:** We followed the standard treatment for calculating distributional discrepancy measures including OT distances used in this work. For vision tasks other than MNIST (on which we directly compute OT distances on the pixel space), **we trained a smaller model (ResNet-18) from scratch on the validation data (owned by the data buyer) and then used its output of the penultimate layer (before the final output layer) as the feature space (on which the OT distance is then computed).**
>
> The train-from-scratch procedure can also be replaced by fine-tuning off-the-shelf pre-trained feature embedders (such as the models pre-trained on ImageNet). Such procedures are more common for language model tasks and less important for vision tasks. For tabular data, generally, no feature embedding is needed and we only normalize the features of different dimensions to the same scale. If the dimension of data is very high (e.g., hundreds), sometimes dimension reduction (e.g., PCA) or feature selection (e.g., RFE, recursive feature elimination) can be applied.

---

> > ### Comment · Reviewer_WNMz · 2023-08-17
> >
> > Thank you for the clarifications.

---

> > > ### Author Response · Authors · 2023-08-20
> > > **Rebuttal period ending–we anticipate your feedback!**
> > >
> > > Dear Reviewer WNMz,
> > >
> > > As the rebuttal/author discussion period is closing, we sincerely look forward to your feedback. The authors are deeply appreciative of your valuable time and efforts spent reviewing this paper and helping us improve it.
> > >
> > > We are compiling the discussions during the review and revising the manuscript to improve its presentation. To better improve its clarity, it would be very much appreciated if you could let us know if our responses and additional results address or partially addresses your concerns on the practicality of its applications and the experiment details and if our explanations are heading in the right direction.
> > >
> > > Please also let us know if there are any further questions or comments about this paper. We strive to consistently improve the paper and it would be our pleasure to have your precious feedback!
> > >
> > > Kind Regards,
> > > Authors of Submission2041

---

### Official Review · Reviewer_Paf5 · 2023-08-01

**Soundness:** 3 good
**Presentation:** 2 fair
**Contribution:** 2 fair
**Rating:** 6
**Confidence:** 2

**Summary:**

The paper considers the data acquisition setting where benefit to model performance from acquiring new streams of training data may be supported by the inspection of limited segments of a candidate corpus, such that one may wish to evaluate the benefit to model performance to support selection from a set of candidate corpi. Such forms of data acquisition evaluation for estimating model performance impact has much prior work, the first claimed novelty in this paper is associated with utilizing forms of optimal transport metrics to improve selecting ratios of data to be served from multiple candidate corpi. This estimate is benchmarked to suggest a resulting improvement to estimates of model performance in comparison to prior work, and then by extending the use of the derived estimates of an ideal ratio to another component of evaluation associated with data scaling laws there is another improvement to estimates of model performance.

**Strengths:**

The abstract claims of significant improvements to computational costs of the application with use of this approach was suggestive of a material contribution but I had difficulty evaluating that further from the rest of the writeup. It is intuitive that for real world application in industry the prioritization of data streams is impactful to bottom line, such that scalable solutions to more easily allocate training compute would be impactful.

**Weaknesses:**

The contributions of the paper associated with incorporation of an optimal transport metric on its own strikes me as much more of an iterative contribution rather than a significant one. I recognize this is common to the field, but I had a hard time getting comfort on the statistical significance of the benchmark findings - even a few more details on experimental setup could have possibly helped. I agree that the figures appear to demonstrate improvement but I don’t think the paper convinced me that it was anything more than suggestive.

**Questions:**

In figure 5 do you have any theories as to why with increasing sample scales the method appears to increasingly underestimate model performance?

**Limitations:**

---

> ### Author Rebuttal · Authors · 2023-08-06
>
> > *“...claims of significant improvements to computational costs of the application with use of this approach was suggestive of a material contribution but I had difficulty evaluating…”* (*Strengths*)
>
> ***TL;DR: Scalability improvements of the proposed framework are on magnitudes.** Methods proposed in this work only require fitting as few as a fixed number of 2 parameters, orders less than existing methods. Figure 3 shows **empirically that the proposed method outperforms the strongest baselines with less than 1/5 computational overhead** of the latters'.*
>
> **Re**: Thanks for pointing out a potential source of confusion in the presentation of this work. **Previous approaches rely on fitting parametric functions as surrogates** (e.g., rational functions [1], or Datamodels [2]), which treat the model and data pipelines as a black box. The accuracy of such methods fundamentally relies on the expressive power of the surrogate functions. **The surrogates are non-informative**–i.e., they predict the performance solely based on the size of data or its composition ratios (how much from each data source) while neglecting the information of the content of data. **The improvement in their accuracy primarily relies on adding more parameters, which require more repetitions of the model training to fit and mitigate the elevated consequence from overfitting.** Under practical limitations on computing resources, the accuracy often needs to be traded off with robustness/reliability (using higher-order nonlinear surrogates may cause large deviations due to overfitting) and is generally unsatisfactory.
>
> **Our methods leverage data distance (measured by the Optimal Transport distance) to incorporate additional information into the prediction of model performance. Our simplest model, OTPP/CS only uses a fixed number of 2 parameters that can be well-fitted through a handful of model training and outperforms large parametric models with the number of parameters quadratic to the number of data sources, demonstrating a substantial advantage over the previous approaches. The sharp contrast highlights the remarkable scalability improvements from the proposed approach.**
>
> Empirical validations are depicted in Figure 3. OTPP/CS converges to a low prediction error (i.e., high accuracy) after 10 times of model training and outperforms non-informative parametric surrogates with up to a quadratic number of parameters even after 50 times of model training.  The scalability improvements are on magnitudes.
>
> The authors acknowledge the responsibility for the presentation of this work and appreciate having the chance to revise the manuscript accordingly for better readability.
>
> > *[1] Tatsunori Hashimoto. Model performance scaling with multiple data sources. In International Conference on Machine Learning, pages 4107–4116. PMLR, 2021.*
>
> > *[2] Ilyas, Andrew, et al. "Datamodels: Predicting predictions from training data.", Proceedings of the 39th International Conference on Machine Learning, 2022.*
>
> ***
>
> > *"...contributions of the paper associated with incorporation of an optimal transport metric ...more of an iterative contribution rather than a significant one."* (*Weaknesses*)
>
> **Re**: We thank the reviewer for the effort to help assess the contribution of this work and for the responsibility held to the community. We would like to take this opportunity to share our understanding of the importance of this work and how it positions in the field. ***[We provide a full response to this question in a separate comment to all reviewers. (R3)]***
>
> ***
> >  *"...significance of the benchmark findings - even a few more details on experimental setup could have possibly helped."* (*Weaknesses*)
>
> **Re**: As part of the effort to improve the presentation of this paper, ***[We provide a full response to this question in the summarization rebuttal to all reviewers. (R1)]***
>
> We apologize for the lack of clarity in experimental details. We are revising our manuscript in light of your valuable feedback.
>
> ***
>
> > *“In figure 5 do you have any theories as to why with increasing sample scales the method appears to increasingly underestimate model performance?”* (*Questions*)
>
> **Re**:  Figure 5 is a qualitative figure that is provided to facilitate the reader’s interpretation of how this framework works. We apologize if it ends up the opposite and adds to the confusion.
> ***[We provide a full response to this question in a separate comment to all reviewers. (R4)]***
>
> We will add more explanations to avoid possible misunderstandings.

---

> > ### Comment · Reviewer_Paf5 · 2023-08-11
> >
> > Acknowledged review of your rebuttal.

---

> > > ### Comment · Reviewer_Paf5 · 2023-08-17
> > >
> > > After further review of comments, submittals, and related literature, I have updated my recommendation to a weak accept. Some further elaborations on the scope of this paper that could make stronger is an expanded discussion on the practicality and logistics for evaluating the OT distance between data sets. Best regards.

---

> > > > ### Author Response · Authors · 2023-08-20
> > > > **The authors appreciate your service!**
> > > >
> > > > Dear Reviewer Paf5,
> > > >
> > > > Once again, the authors express their sincere appreciation for your valuable time and efforts spent reviewing this paper and helping us improve it.
> > > >
> > > > We are revising the manuscript in light of the reviews and compiling the discussions into the manuscript. Your feedback is enormously helpful in improving its presentation and clarity. The authors sincerely appreciate your help in the development of this work as well as your service to the community.
> > > >
> > > > Please also let us know if there are any further questions or comments about this paper. We strive to consistently improve the paper and it would be our pleasure to have your precious feedback!
> > > >
> > > > Kind Regards,
> > > > Authors of Submission2041

---

### Author Rebuttal · Authors · 2023-08-09

### Summary:
**All reviewers recognize the importance of the problem and the conceptual novelty of the proposed framework and original methods.**

Reviewers WNMz and WiXX confirm the **solid development** of this work and its **multi-faceted technical contributions**. Reviewers WNMz and WiXX appreciate the **overall presentation** of this work and acknowledge it as well-written, well-motivated, and well-contextualized with a clear and logical flow of ideas and comprehensive descriptions. Reviewer Paf5 and WiXX acknowledge the **significance of potential impacts** on industrial applications.

**Reviewers share the questions for experiment settings and their practicality.** Reviewer d24K did not include the part of work on predicting model performance for data composed from different combinations of multiple data sources. **Both point to the need for improving the current presentation of the manuscript and including additional discussions.**

***

### In our response, we would like to

- **R1.** Clearly present the experiments conducted in this work, **showcasing their diversity and practicality** and clarifying their settings and considerations.

- **R2.** **Provide additional results of a NEW set of experiments on data selection for automatic speech recognition (ASR)**, a highly practical application in collaboration with industrial practitioners.

- **R3.** Further discuss the **multi-facet contributions** of this work and their **potential impacts** on both industrial and academic research.

- **R4.** Clarify **Figure 5** with additional explanations on its settings and visualization scheme.

***
### R1. Experiments conducted in this work

**a. We validate the proposed pipeline in stylized tasks with standard datasets. On MNIST (simple patterns), IMDB (text data), CIFAR-10 and ImageNet100 (vision tasks)**, we divide the samples into different data sources representing specific data categories, where each source contains only certain classes that are not necessarily exclusive. Then, based on a small amount of pilot data that is considered accessible to the practitioners (i.e., <20% of the target scale), we construct the "projektor" performance prediction tools that visualize the entire performance landscape for any data composition at any data size. We examine the accuracy of these predictions against the actual model performance and compare the error with baselines. Also, we perform data selection based on the predicted performance. We then train the model on the selected data and demonstrate advantageous performance for data selected by "projektor" compared to data selected by baseline methods.

**b. We then examine the practical performance of the proposed methods in real-world instances and extended scenarios.**

Beyond training from scratch, we apply our framework in **data selection tasks for fine-tuning**, which is of high relevance for pre-trained large models that are attracting growing attention. In particular, we implemented “projektor” to select fine-tuning data from the **autonomous driving dataset BDD100K**, the largest open driving video dataset for multi-object tracking (MOT) and segmentation (MOTS) challenges where we use the image frames, for a **Faster-RCNN model pre-trained on COCO**, a large-scale object detection, segmentation, and captioning dataset. For autonomous driving tasks, trained models are often sensitive to diverse weather conditions which limit the visibility of the road. **Thus, we divide the BDD100K data sources into three different challenging visibility categories: daytime, night, or dawn/dusk.**

Similar to previous tasks, we construct our predictors with as few as 1k samples and predict model performance for up to 5k samples.  Visualized in Figure 1(b), **selecting data based on our predictions achieves significantly higher model performance than random selections, and the predictions are highly accurate**–not deviating from the actual accuracy for more than 0.4%, indicating projektor’s practicality in realistic instances and its extended capability for fine-tuning tasks.

Additionally, to model the **effects of data sources with varying data quality**, we included experiments with **added label noise** to a portion of sources (10%-20%). We projected performance from 1K samples onto larger data scales 2-10K and observed (Fig. 4) that the proposed methods **achieve the lowest errors** (MAE scores)–the predicted performance well aligns with the actual performance even in the case of noisy labels, showcasing that the proposed data-distance-based approach **well models the effects of varying data quality and accommodates such tasks with high practical relevance.**

***
### R2. Additional Results: automatic speech recognition (ASR)

For ASR tasks, there are often a number of datasets available, consisting of many audio sources (e.g., accents, recording quality, environmental conditions). **Due to the large scale of samples, training the model each time is considerably resource-intensive.** Finding an ideal combination of data for training on the target tasks can be prohibitively expensive if conducted with manual searches. We consider the case of **fine-tuning a pre-trained Emformer RNN-T model on recordings from LibriSpeech and TEDLIUM-3**, datasets with contrasting speech recording styles–**the former contains clear and stable read speech (scenario 1) and the latter contains spontaneous speech in non-studio environments (scenario 2)**. Our target was to find the best composition of these data sources to improve the fine-tuned performance on both scenarios **with access to limited data (1% of total recordings, ~1hr).**

***(continuing in the comment below)***

---

> ### Author Response · Authors · 2023-08-10
> **R2. ASR(cont’d)/R3. Contributions**
>
> ***(continued from the rebuttal)***
>
> With **almost-precise predictions** of target model performance, **the optimal data selected based on the predictions is nearly as good as the best solution found by intensive grid search. [Results are visualized in the PDF attached to the rebuttal.]** We are compiling these results into a short section in Appendix, highlighting the promising usefulness and broad applications of the proposed methods.
> ***
> ### R3. Additional discussions on technical and conceptual contributions
>
> **On the technical front**, this work proposes an original framework that incorporates the information on data distance into the prediction of model performance, which **enables the construction of highly reliable predictors in a way that is substantially more efficient and scalable**, requiring orders of magnitude less training time. Compared to existing methods based on fitting **non-informative surrogates** for performance prediction–i.e., they predict the performance solely based on the size of data or its composition ratios (how much from each data source) while neglecting the information of the content of data., leveraging the addition of information from data, the resulting performance prediction tools proposed in this work **fundamentally prevent large deviations from overfitting higher-order parametric surrogates**. This property is especially favorable for industrial applications where algorithmic robustness is often a critical concern. For example, Figure 2 gives a direct visualization that the proposed methods achieve a much higher prediction accuracy on extrapolation tasks (predicting model performance on data compositions where training data does not directly cover), **showing a drastic reduction in large prediction errors compared to baselines.**
>
>
> Further, this work extends the state-of-the-art discoveries on scaling laws and adapts them to the context of data selection. A prominent challenge in scaling laws is that model performance for different data scales at different rates. Constructing a universal predictor for data from arbitrary combinations of sources using fixed scaling laws could lead to unsatisfactory results.  This work threads the needle to **organically incorporate the performance predictor with the scaling laws in a parameter-free projection, allowing predicting model performance on any data composition and data quantity in a self-adaptive manner,** automatically fitting scaling parameters for the data based on the prediction tools. This combination enables a totally different type of tools to help inform machine learning practitioners in data selection. In contrast to conventional data acquisition schemes where the algorithm simply acquires data to maximize the utility function and returns the set of collected data, **the tools proposed in this work could visualize the entire landscape of model performance** for any data composition at any data quantity. The practitioner can then make tradeoffs on what type of data to acquire and by how much so that the objectives are best met.
>
>
> **On the conceptual front**, in recent years, distributional discrepancy measures have been popularly used as a proxy of validation performance. A considerable amount of literature has made implicit assumptions that distance implies model performance and subsequently built their works on top of such assumptions. It is largely left unexamined whether, or/and how well, these discrepancy measures correlate with the actual model performance in reality, posing a prominent conceptual gap in research. **This paper is the first to directly use data distance for the prediction of the empirical performance of the model trained on the data**, providing functional relationships that connect them together and successfully developing applications based on these functions. **This work sheds light on a new path that shall provide inspiration for future work on data selection, data valuation, performance prediction, etc.**

---

> ### Author Response · Authors · 2023-08-10
> **R4. Clarifications on Figure 5**
>
> ### R4. Clarifications on Figure 5: Settings and Visualization
>
> **Figure 5 provides a direct visualization of the landscape of prediction errors in model performance** for all data compositions as the scale of data to be predicted grows. We consider the setting of 3 data sources with unlabeled samples from CIFAR-10. **We construct the proposed performance predictors within 1k samples** that are considered accessible to the practitioner. **We then predict model performance for all data compositions and on larger data scales (2-10k samples).**
>
> In Figure 5, X-Y axes represent the proportion of data from data sources #1 and #2 (consequentially, the proportion of data from data source #3 will be 100%-X-Y) and the **prediction errors are visualized in Z-axis** (horizontal plane represents zero error, **red spikes blue dips represent deviations from over-/under-prediction**). As the data scale gets larger, the performance predicted from the initial 1k samples becomes slightly less accurate, but **prediction error remains mostly within 5% even at the scale of 10k data** (10 times larger than the pilot data), **retaining its effectiveness and functionality.**

---

### Author Response · Authors · 2023-08-21
**End of discussion period and thanks note.**

Dear Reviewers and Area Chair,

The authors express their sincere appreciation for your valuable time and efforts spent reviewing this paper and helping us improve it.

Despite the end of author discussions, we strive to consistently improve the paper. Please don't hesitate to let us know anytime if you have any further comments or feedback.

If there are questions from the discussion between the reviewers and the AC, we would be more than glad to know and could take the last few moments today to respond to them and help explain.

Many thanks,
Authors of Submission2041

---

### Decision · Program_Chairs · 2023-09-21

**Decision:**

Accept (poster)

**Comment:**

In the reviewers' words: this paper studies a "well-motivated" and "timely" problem with an approach that is "novel", "interesting" and "well-motivated". It is "well-written" with a "clear set up" and "contextualizes" the work within the literature. Given the unanimous recommendation for acceptance from the four reviewers, I recommend accepting this "valuable contribution".